# FU-DWS: Effective Federated Domain Unlearning via Domain-aware Weight Surgery

## Abstract

Federated Learning (FL) enables distributed clients to collaboratively train machine learning models without sharing raw data, enhancing user privacy. However, stringent data protection regulations, such as the GDPR, mandate the erasure of certain domain-specific knowledge from trained models, raising the critical challenge of federated domain unlearning. Unlike traditional federated unlearning on homogeneous data, federated domain unlearning must remove entire domains' data from a well-generalized model. This difference presents the core challenges of disentangling domain-specific knowledge from shared representations and avoiding excessive forgetting on retained data. To address this challenge, we propose Federated Domain Unlearning via Domain-aware Weight Surgery (FU-DWS). FU-DWS performs "surgical" operations on domain-salient weights to achieve precise and efficient unlearning while preserving the generalization performance on the remaining domains. Specifically, FU-DWS first identifies domain-salient weights by quantifying channel-level domain sensitivity through activation-aware patterns. It then selectively prunes channels strongly associated with the forgetting domain, while freezing stable channels that are critical for the retained domains, thereby accelerating recovery. FU-DWS builds on the Federated Domain Learning framework and evaluates unlearning's ability to support recovery, forgetting, and generalization across different domains in a cross-silo scenario. Comprehensive evaluation against six baselines across three domain-heterogeneous datasets on various backbones demonstrates that FU-DWS significantly outperforms existing methods in both unlearning effectiveness and computational efficiency, while maintaining stronger performance on retained domains.

## 1 Introduction

Federated Learning (FL) enables multiple clients to collaboratively train machine learning models without exchanging raw data, thereby preserving user privacy McMahan et al. (2017). Despite these privacy-preserving advantages, stringent data protection regulations such as the European Union's General Data Protection Regulation (GDPR) European Parliament and Council of the European Union (2016) and the California Consumer Privacy Act (CCPA) California Department of Justice (2020) impose strict requirements on data management and deletion. Under these laws, individuals and organizations have an explicit "right to be forgotten" (RTBF), obligating service providers to erase personal data and any associated information from previously trained models Kalis (2014). To fulfill RTBF mandates, Federated Unlearning (FU) has emerged as an advanced federated mechanism Liu et al. (2023); Jeong et al. (2024); Liu et al. (2021), aiming to remove previously learned knowledge from FL models while maintaining model accuracy on remaining data.

Unlike existing works that concentrate on unlearning data within a homogeneous setting Liu et al. (2021; 2022); Wang et al. (2022); Halimi et al. (2022), our focus is on federated domain unlearning, a more challenging yet practical problem driven by domain heterogeneity in real-world FL Huang et al. (2023); Li et al. (2020). This problem is particularly critical in cross-silo FL scenarios Huang et al. (2023; 2022b); Wang et al. (2024), where clients are typically organizations (e.g., hospitals, financial institutions) with systematically different data distributions, known as domain shifts Chen et al. (2024); Zhang et al. (2023); Zhou et al. (2025). For instance, in federated healthcare, medical imaging datasets collected at different hospitals often exhibit domain shifts driven by differences in

scanner manufacturers and models Kushol et al. (2023), patient demographics Guan et al. (2024), and imaging protocols Kilim et al. (2022). While federated domain generalization techniques have been developed to create robust models under such domain shifts Huang et al. (2023); Zhang et al. (2023), unlearning in such domain shifts is an unresolved issue.

Unlike traditional federated unlearning, as shown in Figure 1 (a), federated domain unlearning introduces unique challenges that extend beyond domain homogeneous data settings. First, removing domain-specific knowledge from a well-generalized global model is inherently difficult, since representations often span across domain boundaries Huang et al. (2023); Zhang et al. (2023). Our empirical results (Section 3.2) show that existing methods Liu et al. (2022; 2021); Halimi et al. (2022); Wang et al. (2022) designed for single-domain cases perform poorly under domain-level heterogeneity. Second, current unlearning approaches Liu et al. (2021; 2022); Wu et al. (2022b); Che et al. (2023) are highly inefficient in domain-unlearning scenarios. Most existing FU methods rely on heuristic optimization and do not provide guarantees specifically tailored to the domain-partitioned FDL setting considered. These methods neither take into account the issue of domain feature overlapping, nor do they consider the requirement for the unlearned model to quickly recovery its retaining performance and domain generalization capabilities. These challenges highlight the need for methods that can selectively remove the influence of a targeted domain from a well-generalized global model while preserving domain-invariant knowledge and ensuring efficient recovery.

To address these challenges, we propose Federated Domain Unlearning via Domain-aware Weight Surgery (`FU-DWS`), which performs "surgical" operations on activation-aware domain-salient weights to enable precise unlearning while preserving high-performance domain generalization. Our approach builds on the key observation that activation differences strongly correlate with domain-specific characteristics. As validated in Section 4.1, these activation patterns provide a reliable signal for identifying model components most critical to a given domain. We refer to these components as activation-aware domain-salient weights, which form the foundation of our design. Based on this insight, `FU-DWS` executes a two-stage surgical procedure: it selectively prunes channels highly sensitive to the forgetting domain to remove its specific knowledge, while simultaneously freezing stable channels that capture domain-invariant information to support rapid and reliable recovery on the retained domains. Our experiments demonstrate that `FU-DWS` achieves both effective and resource-efficient federated domain unlearning.

The key contributions of this work are summarized as follows:
• To the best of our knowledge, we are **the first** to systematically address the challenges of federated domain unlearning in terms of *recovery, forgetting, generalization*. To this end, we propose `FU-DWS`, a novel algorithm that precisely removes domain knowledge while minimizing collateral forgetting and efficiently restoring model performance.
• We design a two-stage surgical operation on a well-generalized model, leveraging activation-aware domain-salient weights to achieve precise and efficient unlearning while preserving strong domain generalization.
• We conduct extensive experiments on three heterogeneous-domain datasets, demonstrating the superiority of `FU-DWS` over six baselines. Moreover, we validate its effectiveness across multiple backbone architectures, including both CNNs and Vision Transformers (ViTs), establishing its wide applicability and robustness. Our method achieves state-of-the-art performance and delivers up to $25\times$ higher efficiency than retrain.

## 2 RELATED WORK

**Federated learning with cross-silo domain heterogeneity.** In practical cross-silo FL settings, data heterogeneity often manifests as structural domain shifts Nguyen et al. (2025); Wang et al. (2025) between organizations (e.g., hospitals), going beyond simple client drift Kairouz et al. (2019); Li et al. (2021). To address this, federated domain learning has primarily followed two main approaches: *Prototype Learning* abstracts domain-specific features into generalizable, domain-invariant prototypes to stabilize training across varied domains Huang et al. (2023; 2022b); Wang et al. (2024). These methods enhance model robustness but focus on performance optimization rather than the targeted removal of domain knowledge. *Domain Adaptation and Generalization* techniques, on the other hand, aim to align or re-weight feature representations across silos, often through adversarial training or principled feature alignment Huang et al. (2022a); Zhang et al. (2023); Jiang et al. (2023); Le et al.

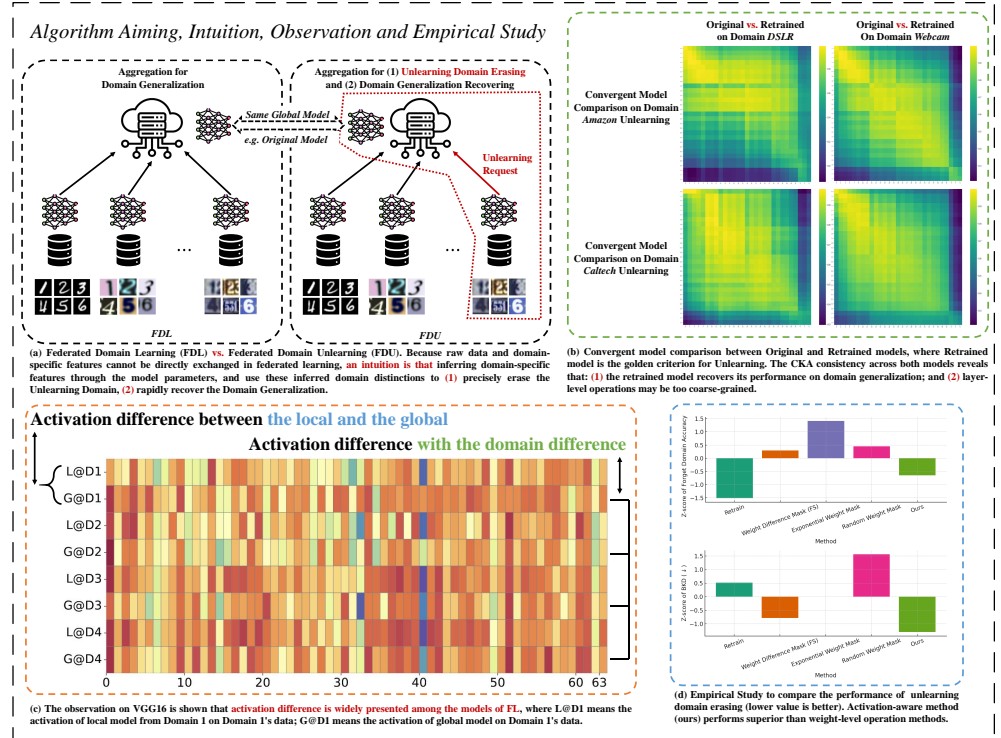

Figure 1: Design Framework of FU-DWS. (a) shows the Algorithm Aiming and Intuition of method design. (b) and (c) presents the observation on model surgery signal between layer-level and channel-level for domain unlearning. (d) shows the empirical study on different granularities domain surgery.

(2024). In this work, we discuss Federated Domain Unlearning on these cross-silo scenarios, which enforces precise forgetting of domain-specific information within well-generalized models.

**Federated Unlearning** Existing federated unlearning research has primarily addressed client-level and class-level unlearning within homogeneous domains. These approaches can be categorized into three groups: *1.Retrain unlearning* transplants exact unlearning methods from centralized learning without leveraging the partition-aggregation framework, limiting efficiency gains in federated settings. Rapid retrain methods Liu et al. (2021; 2022) attempt to accelerate retraining by approximating gradients using stored historical updates; *2.Reverse Unlearning* removes learned data through gradient operations, including loss maximization Halimi et al. (2022) and gradient manipulation Wu et al. (2022b); Che et al. (2023). These techniques approximate loss functions on remaining data by calculating inverse Hessian matrices or injecting noise to smooth local gradients during aggregation; *3.Alternative Approaches* include knowledge distillation Wu et al. (2022a), scaled gradients Gao et al. (2024), and channel pruning for CNNs Wang et al. (2022). These methods generally have weaker theoretical foundations and may introduce greater privacy vulnerabilities. While effective in single-domain scenarios, the applicability of these methods remains unexplored in heterogeneous multi-domain federated learning environments where domain boundaries create unique challenges for selective knowledge removal.

## 3 CHALLENGES ON DOMAIN UNLEARNING IN FL

### 3.1 FEDERATED DOMAIN UNLEARNING

**Cross-silo setting.** We consider a *cross-silo* federation with a finite set of domains (silos) $\mathcal{D} = \{d_1, \ldots, d_M\}$. Following the Existing works setting Nguyen et al. (2025); Wang et al. (2025), Each client $k$ belongs to exactly one domain $d(k) \in \mathcal{D}$; the client set of domain $d$ is $\mathcal{C}_d = \{k : d(k) = d\}$. All clients in the same silo share the *same* underlying distribution $P_d(x, y)$ (same label space $\mathcal{Y}$, domain-specific feature marginal), and we write their dataset generically as $\mathcal{D}_d$; we use $\mathcal{D}_k$ only when emphasizing the holder. A standard cross-silo objective learns a global model $f_\theta : \mathcal{X} \to \mathcal{Y}$ over

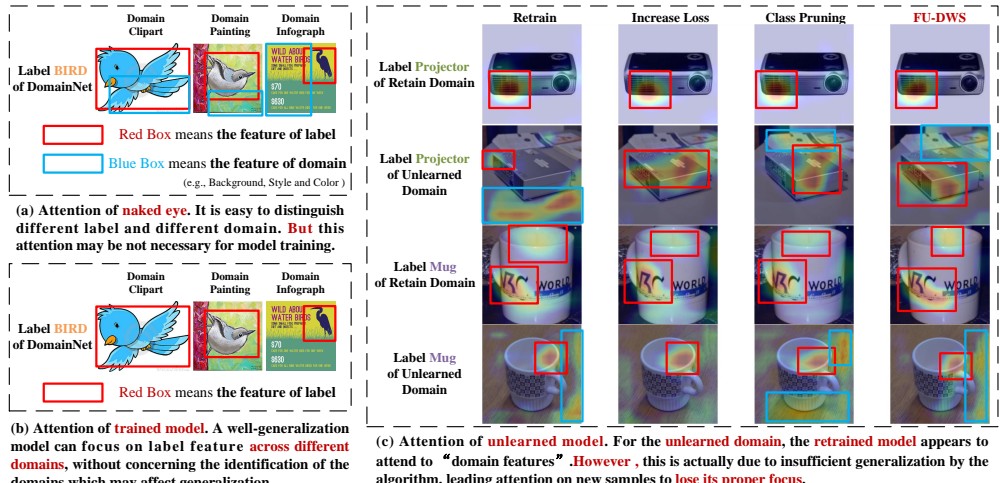

Figure 2: Attention map to observe overlapping domain representations of global model. (a) and (b) compare human visual classification with model classification, emphasizing that the well-generalized model attends primarily to label-specific features. (c) shows that **Retrain** (the **Golden** baseline) yields attention maps in the unlearning domain which do not restrict their focus to label features.

training domains $\mathcal{D}_{\text{train}}$ by

$$\min_{\theta} \sum_{d \in \mathcal{D}_{\text{train}}} w_d \, \mathbb{E}_{(x,y) \sim \mathcal{D}_d} \big[ \mathcal{L}\big(f_\theta(x), y\big) \big] , \tag{1}$$

where $w_d \geq 0$ are aggregation weights.

**Federated domain unlearning.** Given a model $f_\theta$ trained on $\mathcal{D}_{\text{train}} = \mathcal{D}_{\text{retain}} \cup \{d_f\}$, an unlearning request specifies a *forget domain* $d_f$. All clients in $\mathcal{C}_{d_f}$ withdraw and, after the request, no samples or gradients from $\mathcal{D}_{d_f}$ are accessed. The goal is to transform $f_\theta$ into $f_{\theta'}$ that (i) behaves as if it had *never* seen $\mathcal{D}_{d_f}$ and (ii) preserves utility on the retained set $\mathcal{D}_{\text{retain}}$:

$$\min_{\theta'} \sum_{d \in \mathcal{D}_{\text{retain}}} w_d \, \mathbb{E}_{(x,y) \sim \mathcal{D}_d} \big[ \mathcal{L}\big(f_{\theta'}(x), y\big) \big] \quad \text{s.t.} \quad f_{\theta'} \approx f_{\theta \setminus d_f}, \tag{2}$$

where $f_{\theta \setminus d_f}$ denotes the retrain on $\mathcal{D}_{\text{retain}}$ alone, which is infeasible in the real world. In practice we certify "$\approx$" via surrogate behavioral checks without retraining.

*Takeaways.* Federated unlearning methods are only focus on privacy data erasing on Forget set and model recovery performance on Retain set. Federated domain unlearning methods need to pay extra attention to maintaining the generalization ability of the model, even for the test performance of the Forget set.

## 3.2 CHALLENGES FOR DOMAIN UNLEARNING IN FL

**Overlapping Domain Representations of global model.** Unlike sample or class unlearning, domain-level unlearning faces a critical challenge: *precisely extracting and removing domain-specific knowledge from a well-generalized federated model is difficult*. To illustrate this dilemma of overlapping representations, we analyze the contrast between human and model classification by examining the characteristics revealed in attention maps. As depicted in Figure 2(a), when presented with a cross-domain image classification task, the human eye can readily distinguish between *label features* (the object of interest) and *domain features* (e.g., style, background). An ideally well-generalized model should emulate this focus, as shown in Figure 2(b), by concentrating its attention sharply on label features while ignoring domain-specific distractions. However, Figure 2(c) reveals a critical phenomenon when we visualize the attention of existing unlearning methods on samples from the unlearned domain that were seen during training. For the golden baseline, Retrain, the model's attention becomes noticeably diffuse: it no longer locks onto the label feature as it does for retained domain samples. This apparent shift towards domain features is actually a consequence of the model forgetting the sample, leading the once well-generalized model to lose its precise focus. This phenomenon crystallizes the core challenge of federated domain unlearning: *one must simultaneously*

*protect performance on retained domains, forget the unlearned domain, and preserve the federated domain learning's fundamental goal of generalization.* While this trade-off is implicitly handled by the brute-force Retrain approach, it poses a formidable difficulty for approximate unlearning methods that aim to achieve both efficiency and effectiveness by modifying a pre-trained model directly.

**Computational and Communication Inefficiency.** A significant challenge in federated domain unlearning is its inherent inefficiency in both computation and communication compared to unlearning in homogeneous settings. This inefficiency stems from two compounding factors. First, it is well-documented that statistical heterogeneity (i.e., non-IID data or domain skew) alone slows and destabilizes federated optimization, requiring more communication rounds and local computation to reach a target performance level even before unlearning is considered Li et al. (2019); Kairouz et al. (2019); Li et al. (2020). Second, the act of unlearning itself introduces a significant overhead, whether through partial retraining or influence-based adjustments, which add time, memory, and storage costs Liu et al. (2021; 2023). These factors are exacerbated in the context of domain unlearning. Because domain-specific knowledge is deeply intertwined with shared, domain-invariant representations across multiple layers, isolating and removing one domain's influence without causing collateral damage demands more complex and extensive corrective fine-tuning. This directly translates to a greater number of recovery rounds. Our evaluations show that existing methods like FedEraser Liu et al. (2021) require up to 3× more communication rounds to recover performance on retained domains compared to single-domain unlearning scenarios.

## 4 THE PROPOSED METHOD

This section introduce the construction of our core methodology `FU-DWS`. We first introduces its cornerstone observation in Section 4.1: neural network activations, particularly at the channel level, are domain-dependent. To substantiate this hypothesis, the section provides critical empirical evidence by visualizing significant activation differences between local and global models and across various domains. Furthermore, it translates this observation into an operable signal through channel ablation experiments, which confirm that activation magnitude serves as a reliable proxy for domain importance. Building on this validated insight, Section 4.2 details how this principle is operated into the `FU-DWS` framework, an activation-driven "surgical" unlearning procedure. Figure 3 shows the pipeline of `FU-DWS`: *(1) detecting activation-aware domain-salient weights → (2) pruning on the global model → (3) per–domain freezing → (4) local updating & resetting → (5) aggregation.* No raw data from $D_u$ is used after the request.

### 4.1 ACTIVATION-BASED DOMAIN SENSITIVITY

We first illustrate the rationale behind exploring Activation-based Salient Weights mapping for domain sensitivity. Recent advances Lin et al. (2024); Ben-David et al. (2006); Li et al. (2017) suggest that neural networks develop specialized internal representations for different data distributions. We deliberately decouple this clue from the quantization context and hypothesize a distributional statement: *per–layer, per–channel activations of a multi–domain global model are domain–dependent and can disagree with client–local statistics.* Specifically, when exposed to domain-specific data patterns, neural networks allocate specific channels to capture domain-unique features, while other channels focus on domain-invariant features. This internal organization enables effective domain adaptation even within a unified model structure. The visualizations reveal distinct patterns. Figure 1 (b)-(d) summarizes the observation and empirical evidence, from intuition to observation.

Specifically, in federated setting, to avoid increasing communication overhead and privacy leakage risks, one **intuition** is to directly surgery on the model and hope to isolate the unique domain features within it. Among them, the three granularities of layer-level, channel-level and weight-level represent the main signal operation levels that we consider. To verify the availability of our intuition, we analyzed the different granularities operation on model surgery. Compared between Figure 1 (b) and (c), we observed that directly considering the operation granularity at the layer level was too coarse, and there was no significant difference between the Retrain and the original model. However, the Activation-aware Saliency enabled the channel-level operations to be feasible. And we conducted empirical study on more granular scale. In Figure 1 (d), we compared more granular weight-level operations, including the modified FedSalUn based on SalUn (Fan et al., 2023),

Figure 3: The framework of our proposed method. In (a) FU-DWS detects the activation-aware domain-salient weights; and in (b), FU-DWS use these weights to realize surgical domain unlearning.

exponential distribution weighted, and random mask mechanisms. We found that the channel-level operations were more effective for federated domain unlearning.

To ask whether activations can surface domain differences in FL, for each layer $l$ and channel $c$, we compute the mean activation of the *local* domain-specific model and the *global* well-generalized model on the *same* domain $D_k$ (spatially and mini-batch averaged). We visualize these per–channel statistics as a heatmap with a shared, per-layer normalization so that colors are comparable across rows: as shown in Figure 1 (c), for every layer we pool all rows $\{L@D_k, G@D_k \text{ over } k\}$ to estimate a single $(\mu_l, \sigma_l)$ and plot $z$-scores. We have observed that (i) comparing the two rows $L@D_k$ vs. $G@D_k$ yields a *local–global* difference on domain $D_k$ (domain-specific local model vs. well-generalized global model); (ii) comparing $G@D_i$ across different $i$ yields a *cross-domain (global)* difference: the global model's activation baseline shifting across domains. This setup respects known facts: activation statistics drift across domains (*feature shift*), while deeper layers tend to specialize more strongly than shallow ones Li et al. (2021). So a minority of channels may carry most domain-specific characters. This pattern motivates two working hypotheses that guide our method: **(1)** Activation differences may trace domain-specific components. Channels repeatedly exhibiting large local–global or cross-domain discrepancies are plausible carriers of domain-specific signal rather than domain-invariant features; **(2)** Contrasting local with global activations provides an indirect handle for a domain-specific local model to "sense" knowledge present in other domains via the global model, without collapsing domain-invariant structure.

**From Differences to Operable Signals.**

**Take-away and bridge to the method.** Figure 1 jointly support our premise: *activation differences (local–global and cross–domain) provide an operative proxy for domain specificity*. This motivates a *Activation-aware Domain-salient Weights Detector* in Section 4: we derive a pruning mask $M_p$ by selecting channels with large domain sensitivity and a freezing mask $M_f$ by selecting channels to aware potential other domain knowledge.

### 4.2 ACTIVATION-DRIVEN SURGICAL UNLEARNING

**Channel importance and Activation-aware Domain-salient Weights Detector.** Motivated by Section 4.1, our Activation-aware Domain-salient Weights Detector builds a binary mask for each layer by thresholding a channel-wise saliency tensor. The saliency for layer $l$ and output channel (or neuron) $c$ on client $k$ is

$$S_{l,c}(\theta^{(k)}, \mathcal{D}_k) = \frac{1}{|\mathcal{D}_k|} \sum_{x \in \mathcal{D}_k} \text{mean}_{h,w} \left| A_{l,c}(x; \theta^{(k)}) \right|, \quad (3)$$

where $A_{l,c}(x; \theta^{(k)})$ is the activation map produced by channel $c$ for input $x$ under model parameters $\theta$ and the spatial mean is replaced by a feature-dimension mean for linear layers. In empirical, each forward pass registers hooks on every Conv2d/Linear layer, accumulates the per-channel mean absolute activations across batches, and averages them over all calibration samples to obtain $S_{l,c}$. We then apply layer-wise min–max normalization so that every layer's scores lie in $[0, 1]$, compute a importance-aware score for the target domain/model variant, and finally invoke the shared mask-selection operator to produce a binary channel mask. Only the input score tensor changes between our detectors; the masking rule itself is identical.

**Surgical Domain Unlearning on erasing: local pruning for domain–specific unlearning.** Given the received global model on client $k$, we form the pruning scores by

$$\Gamma_{l,c}^{(p)} = S_{l,c}\big(\theta^{(k)}, \mathcal{D}_k\big), \tag{4}$$

and obtain the *pruning* mask by a fixed threshold $\tau_p$:

$$M_{l,c}^{(p)} = \mathbf{1}\Big[\Gamma_{l,c}^{(p)} > \tau_p\Big]. \tag{5}$$

Channels with $M_{l,c}^{(p)} = 0$ are pruned (gated) from the received global model:

$$\tilde{\theta}_{l,c}^{(k)} = M_{l,c}^{(p)} \odot \theta_{l,c}. \tag{6}$$

**Surgical Domain Unlearning on maintaining: selective freezing for rapid recovery.** To stabilize shared features during local recovery, client $k$ measures the *temporal* activation stability between the current and previous global models:

$$\Gamma_{l,c}^{(f)} = \delta_{l,c} = \big|S_{l,c}\big(\theta^{(k)}, \mathcal{D}_k\big) - S_{l,c}\big(\theta_{\text{prev}}, \mathcal{D}_k\big)\big|. \tag{7}$$

A *freezing* mask is then produced with threshold $\tau_f$:

$$M_{l,c}^{(f)} = \mathbf{1}\Big[\Gamma_{l,c}^{(f)} > \tau_f\Big], \tag{8}$$

so that channels with *small* temporal change ($\delta_{l,c} \leq \tau_f$, i.e., $M_{l,c}^{(f)} = 0$) are frozen during local optimization. Local updates on client $k$ are applied only to channels allowed by $M^{(f)}$:

$$\tilde{\theta}_{l,c}^{(t+1)} = \tilde{\theta}_{l,c}^{(t)} - \eta \, \nabla\mathcal{L}\big(\tilde{\theta}^{(t)}\big)_{l,c} \odot M_{l,c}^{(f)}. \tag{9}$$

**Putting it together (`FU-DWS`).** After the forgotten domain $\mathcal{C}_F$ issues the request, the server distributes $(\theta, \theta_{\text{prev}})$ to retained clients. Each retained client $k$ computes $\Gamma^{(p)}$ and $\Gamma^{(f)}$ with the *same scoring operator* $S_{l,c}(\cdot, \mathcal{D}_k)$, produces $M^{(p)}$ and $M^{(f)}$ by the *same* selection rule, prunes $\theta$ with $M^{(p)}$, and performs a short selective fine–tuning with gradient masking by $M^{(f)}$. Batch–Norm resetting and local–head replacement are applied before fine–tuning; these implementation details are standard and deferred to the Appendix A. Finally, the server aggregates updated models to obtain the unlearned global model. Algorithm 1 is also shown in Appendix A

## 5 EXPERIMENTS

The main text presents our experimental analysis, covering the basic setup, evaluation metrics, and the effectiveness and efficiency of the unlearning process. Supplementary materials are available in Appendix A, which includes the detailed experimental setup (Appendix A.1), extensive experiments (Appendix A.2), ablation studies on different unlearning granularities (i.e., layer, channel, and weight) with a focused discussion on FedSalUn (Appendix A.3 and A.4), a hyperparameter sensitivity analysis (Appendix A.5), and the results under shared-domain setting (Appendix A.6).

### 5.1 SETUP

**Datasets, Models and Baselines.** We conducted the experiments on **Domain-Digits** Hull (1994); LeCun et al. (1998); Netzer et al. (2011); Roy et al. (2018); Ganin & Lempitsky (2015)with the *CNN*; **Office-Caltech-10** Gong et al. (2012) with *VGG16* for unlearning on the training and *ResNet50 & ViT* for unlearning on the fine-tuning; and **PACS** Li et al. (2017) with VGG16. We compared the performance with Retrain (RE), Rapid Retrain (RR) Liu et al. (2022), FedEraser (FE) Liu et al. (2021), Increase Loss (IL) Halimi et al. (2022), Class-Discriminative Pruning (CP) Wang et al. (2022) and FedSalUn (FS; we replicated and adjusted this method) Fan et al. (2023).

**Validation Metrics.** We use accuracy ($Acc$) and backdoor attack success rate ($BKD$) as validation methods for forgetting effectiveness Tam et al. (2024). Accuracy, akin to Retrain, serves as the gold-standard metric for evaluating all methods. We use $Acc_{domain}$ to show the accuracy of such domain

Table 1: Average performance (%) on last 5 epochs of various datasets.

| Methods | Retain Performance | | Forget Performance | | Test Performance | | | Overall (↑) |
|---|---|---|---|---|---|---|---|---|
| | $Acc_1(\uparrow)$ | $Acc_2(\uparrow)$ | $Acc_F(\downarrow)$ | $BKD(\downarrow)$ | $Acc_1(\uparrow)$ | $Acc_2(\uparrow)$ | $Acc_F(\uparrow)$ | Performance |
| Domain-Digits & Unlearning Domain = MNIST | | | | | | | | |
| RE | 94.88 | 99.05 | 53.67 | 0.29 | 82.15 | 97.40 | 97.4 | - |
| RR Liu et al. (2022) | -13.19 | -1.40 | -1.29 | +0.47 | -13.66 | -1.56 | -2.73 | -3.47 |
| FE Liu et al. (2021) | -8.37 | -2.19 | -0.59 | +0.11 | -3.98 | -2.23 | -0.97 | -1.94 |
| IL Halimi et al. (2022) | +4.20 | +0.82 | +0.92 | +0.04 | +1.50 | +0.58 | +0.39 | +0.95 |
| CP Wang et al. (2022) | +2.24 | +0.76 | +0.52 | -0.01 | +0.04 | +0.07 | +0.48 | +0.55 |
| FS Fan et al. (2023) | +3.81 | +0.72 | +1.02 | +0.04 | +1.10 | +0.70 | +0.54 | +0.85 |
| Ours | +4.38 | +0.76 | +0.45 | -0.07 | +1.93 | +0.94 | +0.71 | **+1.15** |
| Office-Caltech-10 & Unlearning Domain = Caltech | | | | | | | | |
| RE | 47.96 | 74.21 | 19.75 | 1.90 | 44.06 | 75.56 | 32.18 | - |
| RR Liu et al. (2022) | -11.59 | -31.40 | +0.11 | +22.05 | -8.44 | -44.44 | -8.44 | -3.56 |
| FE Liu et al. (2021) | -2.32 | -27.60 | -1.33 | -0.34 | +0.18 | -31.67 | +0.18 | -1.81 |
| IL Halimi et al. (2022) | +34.31 | +15.87 | +16.33 | +1.90 | +15.82 | +7.77 | +15.82 | +1.48 |
| CP Wang et al. (2022) | +2.43 | +11.73 | +3.14 | +1.32 | +6.58 | +4.45 | +6.58 | +0.35 |
| FS Fan et al. (2023) | +12.32 | +14.05 | +29.15 | +18.63 | +14.13 | +10.55 | +14.13 | -0.17 |
| Ours | +35.59 | +16.70 | +11.51 | +0.00 | +14.85 | +5.00 | +14.85 | **+1.65** |
| PACS & Unlearning Domain = Art_painting | | | | | | | | |
| RE | 79.75 | 87.72 | 21.22 | 5.70 | 71.75 | 80.81 | 73.42 | - |
| RR Liu et al. (2022) | -20.65 | -37.80 | -7.26 | -2.82 | -11.34 | -38.73 | -12.12 | -1.73 |
| FE Liu et al. (2021) | -67.82 | -69.29 | -16.21 | -5.70 | -59.88 | -56.20 | -25.93 | -4.47 |
| IL Halimi et al. (2022) | -3.66 | +5.64 | +8.36 | +4.87 | +1.13 | +5.04 | -5.21 | 0.08 |
| CP Wang et al. (2022) | +0.21 | +9.70 | +4.06 | +1.41 | +3.50 | +5.89 | +0.97 | +0.58 |
| FS Fan et al. (2023) | +7.05 | +2.68 | +23.42 | +7.98 | +6.89 | +1.53 | +2.29 | -0.01 |
| Ours | +13.08 | +6.10 | +5.36 | +3.50 | +9.38 | +5.77 | +0.00 | **+0.79** |

like $Acc_A$ for Amazon, and use $Acc_F$ to perform the certain unlearning domain. $BKD$ is employed as a supplementary tool to analyze whether clients have genuinely forgotten the target images. Due to some baselines overly emphasize forgetting, resulting in poor retain and test performance (e.g., FE and RR), or focus on quickly restoring retain performance while neglecting forgetting (e.g., IL, CP, FS), we define a more intuitive evaluation metric called *Overall Performance*. To produce this single, interpretable *Overall Performance* score that balances recovery (retain), generalization (test) and forgetting, we proceed as follows. Let $Acc_{retain}$ denote the model's recovery ability measured at the specified checkpoint(s) (higher is better), and let $Acc_{test}$ denote the model's post-recovery generalization on held-out data (higher is better). Let $Acc_{forget}$ and $BKD$ denote the remaining accuracy on the to-be-forgotten domain and the backdoor attack success rate respectively (both lower values indicate stronger unlearning). We first combine the two forgetting indicators into a single forgetting measure $F = Acc_{forget} + BKD$ so that larger $F$ means worse forgetting. To remove scale differences between these quantities we convert each into a z-score computed across the set of compared methods/variants (i.e., the mean and standard deviation are taken over the methods we evaluate). Concretely, with a small stabilizer $\varepsilon > 0$ to avoid division by zero,

$$z_R = \frac{Acc_{retain} - \mu_{all}}{\sigma_{all} + \varepsilon}, \qquad z_G = \frac{Acc_{test} - \mu_{all}}{\sigma_{all} + \varepsilon}, \qquad z_F = \frac{F - \mu_{all}}{\sigma_{all} + \varepsilon}, \qquad (10)$$

where $(\mu_{all}, \sigma_{all})$ is the empirical mean and standard deviation of $Acc_{retain}$, $Acc_{test}$ and $F$ across the compared methods. The purpose of this normalization is to enhance the impact of the $BKD$ small values. This Overall Performance score used in tables and plots is then

$$\text{Overall} = \sum z_R/n_R + \sum z_G/n_G - \sum z_F/n_F. \qquad (11)$$

where $n_R$, $n_G$ and $n_F$ are the number of the corresponding set. Under this definition a larger Overall is better: methods that simultaneously increase recovery ($z_R$) and generalization ($z_G$) while reducing the forgetting measure ($z_F$) receive higher scores. Note this is a standardized, heuristic aggregation intended to aid comparison; all underlying raw metrics ($Acc_{retain}$, $Acc_{test}$, $Acc_{forget}$, $BKD$) must still be reported because the scalar Overall can hide important trade-offs.

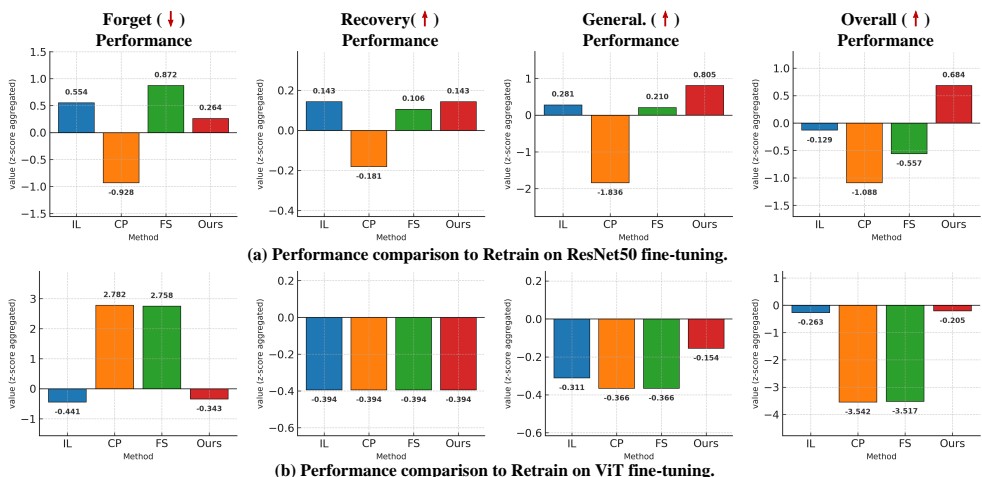

(a) Performance comparison to Retrain on ResNet50 fine-tuning.

(b) Performance comparison to Retrain on ViT fine-tuning.

Figure 4: Effectiveness comparison to RE's performance on ResNet50 and ViT fine-tuning. FU-DWS shows superior performance across various metrics.

## 5.2 UNLEARNING EFFECTIVENESS

**(1) Unlearned on the Training.** As demonstrated in Table 1, our method, `FU-DWS`, consistently achieves the best *Overall* performance across all representative datasets, securing top scores of +1.15 (+0.20 to Top-2), +1.65 (+0.17 to Top-2), and +0.79 (+0.21 to Top-2) in the three distinct unlearning scenarios. Given that we have normalized Overall Performance, the actual gap is even larger. This highlights its superior ability to balance the competing demands of unlearning. A detailed comparison reveals the clear trade-offs made by baseline methods. For instance, methods like **RR** and **FE** often cause catastrophic forgetting on retained domains, with **FE**'s retain accuracy dropping by as much as 67.82% in the PACS dataset. Conversely, methods such as **IL** and **FS** fail to effectively unlearn, evidenced by their high forget-domain accuracy (*AccF*); on the Office-Caltech-10 dataset, their *AccF* reached +16.33% and +29.15% respectively, indicating poor knowledge removal. Other methods like **CP** offer a mediocre compromise without excelling in any particular metric. In stark contrast, `FU-DWS` successfully navigates these trade-offs. On the challenging PACS dataset, it not only boosts retained-domain performance (e.g., +13.08%) but also maintains strong forgetting metrics, achieving the highest *Overall* score. This superior balance validates our core hypothesis that channel activations provide a sharper, more effective signal for domain-specific knowledge removal than the gradient-based or heuristic approaches used by competing methods.

**(2) Unlearned on the Fine-tuning.** The pattern shown in Figure 4 persists under post-hoc fine-tuning on ResNet50/ViT: our method maintains strong *Recovery* and *Generalization* while retaining strict *Forget* control, yielding the highest or near-highest *Overall* across backbones; **CP** remains competitive on forgetting but continues to trade off retain/test, and **FS** trails on forgetting despite decent retain. These results further substantiate that *activation-patterns* encode domain sensitivity that can be converted into actionable masks, enabling precise domain removal without sacrificing shared features.

## 5.3 UNLEARNING EFFICIENCY

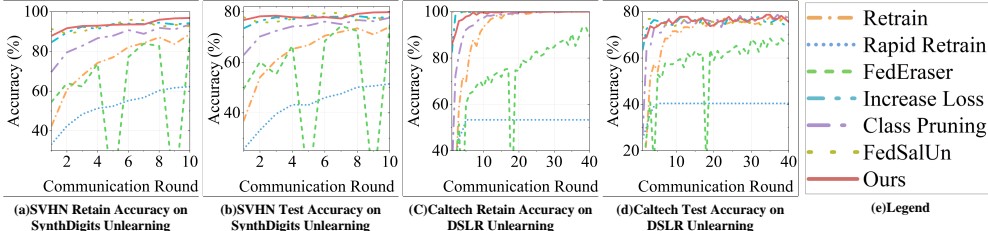

(a)SVHN Retain Accuracy on SynthDigits Unlearning
(b)SVHN Test Accuracy on SynthDigits Unlearning
(C)Caltech Retain Accuracy on DSLR Unlearning
(d)Caltech Test Accuracy on DSLR Unlearning
(e)Legend

Figure 5: Speedup Comparison on Domain SVHN (a,b) of Domain-Digits and Caltech (c,d) of Office-Caltech-10.

Table 2: Retain Performance Recovery Speedup on Office-Caltech-10. We denote the communication round of each method to approach the target accuracy as "R#", the corresponding convergence speedup relative to **Ours' Accuracy in the first rounds** as "S↑".

| Methods | Unlearning Domain = Caltech | | | | | | Unlearning Domain = Webcam | | | | | |
| | Amazon | | DSLR | | Webcam | | Amazon | | Caltech | | Webcam | |
| | R# | S↑ | R# | S↑ | R# | S↑ | R# | S↑ | R# | S↑ | R# | S↑ |
| --- | --- | --- | --- | --- | --- | --- | --- | --- | --- | --- | --- | --- |
| Target | 69.84% | | 76.86% | | 91.56% | | 74.02% | | 75.95% | | 70.25% | |
| RE | >40 | 0.03× | 25 | 0.04× | 8 | 0.13× | 5 | 0.02× | **1** | **1×** | 8 | 0.13× |
| RR Liu et al. (2022) | >40 | 0.03× | >40 | 0.03× | >40 | 0.03× | >40 | 0.03× | >40 | 0.03× | >40 | 0.03× |
| FE Liu et al. (2021) | >40 | 0.03× | >40 | 0.03× | >40 | 0.03× | 13 | 0.08× | >40 | 0.03× | >40 | 0.03× |
| IL Halimi et al. (2022) | 3 | 0.33× | 2 | 0.5× | 2 | 0.5× | **1** | **1×** | **1** | **1×** | **1** | **1×** |
| CP Wang et al. (2022) | 3 | 0.33× | 5 | 0.2× | 8 | 0.13× | 2 | 0.5× | **1** | **1×** | 4 | 0.25× |
| FS Fan et al. (2023) | >40 | 0.03× | **1** | **1×** | **1** | **1×** | **1** | **1×** | **1** | **1×** | **1** | **1×** |
| Ours | **1** | **1×** | **1** | **1×** | **1** | **1×** | **1** | **1×** | **1** | **1×** | **1** | **1×** |

We further evaluate unlearning efficiency through speed-up comparisons depicted in Figure 5 and recovery times summarized in Table 2. Figure 5 clearly illustrates that **FE** experiences frequent training divergences, significantly compromising stability across multiple datasets and domains. **RR**, while stable on smaller domains, faces practical issues such as out-of-memory errors on larger datasets due to computationally intensive second-order gradient calculations, limiting its real-world applicability. Additionally, **CP** consistently exhibits slower recovery speeds.

Table 2 emphasizes our method's superior recovery efficiency, achieving the fastest performance across both domain-unlearning settings, even surpassing Retrain by up to 25 times in certain configurations. Although **FS** shows comparably fast recovery, its incomplete forgetting (as discussed in Section 5.2) renders its efficiency less meaningful in practical unlearning contexts.

# 6 CONCLUSION

This paper proposes `FU-DWS`, a federated domain unlearning framework that removes domain-specific knowledge via activation-guided weight surgery. By identifying sensitive channels through local activations, our method enables precise pruning and selective recovery without costly retraining. `FU-DWS` preserves utility on retained domains while ensuring effective forgetting, requiring only local computations and maintaining data privacy. Experiments show superior recovery & forget & generalization performance on various domains and up to $25\times$ faster recovery in certain configurations compared to existing baselines, making it a practical solution for privacy-preserving federated learning.

## ETHICS STATEMENT

This work proposes FU-DWS, a method for selectively removing domain-level learned representations in federated learning cross-silo scenarios to support compliance and data-governance requirements. Our experiments use only public image benchmarks (Domain-Digits Hull (1994); LeCun et al. (1998); Netzer et al. (2011); Roy et al. (2018); Ganin & Lempitsky (2015), Office-Caltech-10 Gong et al. (2012), PACS Li et al. (2017)). The method is intended to assist in meeting compliance goals such as those in Article 17 of the EU GDPR (the "right to be forgotten") European Parliament and Council of the European Union (2016) and California Consumer Privacy Act (CCPA) California Department of Justice (2020), but it does not constitute legal advice; actual production deployment must be carried out under the supervision of the data controller and legal counsel.

## REPRODUCIBILITY STATEMENT

Code and scripts are provided in the supplementary materials to replicate the empirical results.

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

# A    Technical Appendices and Supplementary

In this chapter, we aim to supplement our motivation and empirical findings, providing more ideas for future research. In Section A.1, we introduce the detailed experiments settings. In Section A.2, we introduce the extensive experiments to verify the effectiveness and robustness of FU-DWS. In Section A.3, we discuss the possibilities of performing weight operations at the layer-level, channel-level, and weight-level, as well as the insights gained from our observations. In Section A.4, we explain why our method employs layer operations based on resetting BatchNorm and the Classifier, and show the ablation test of FU-DWS. In Section A.5, we examine the selection of hyperparameters and the potential for personalized design with Hyper-parameters sensitvety test. In section A.7, we reveal our LLM Usage. The overall procedure is detailed in Algorithm 1.

---

**Algorithm 1** Client-Centric Activation-Based Federated Unlearning

---

**Input:** Current global model $\theta$, previous global model $\theta_{prev}$, local models $\{\theta^{(k)}\}_{k \in \mathcal{C}_R}$, local datasets $\{\mathcal{D}_k\}_{k \in \mathcal{C}_R}$, client to forget $\mathcal{C}_F$.
**Parameters:** Pruning threshold $\tau_p$, freezing threshold $\tau_f$, local fine-tuning epochs $E_{ft}$.
**Output:** Unlearned global model $\tilde{\theta}$.

1: Initialize an empty list for client-updated models.
2: **for all** clients $k \in \mathcal{C}_R$ **in parallel do**
3:      Receive models $\theta$, $\theta_{prev}$ from server.
4:      Load local model $\theta^{(k)}$ and dataset $\mathcal{D}_k$.
                                           ▷ Channel Sensitivity Computation
5:      **for all** layers $l$, channels $c$ **do**
6:          Compute $S_{l,c}^{(local)} = S_{l,c}(\theta^{(k)}, \mathcal{D}_k)$ using Eq. (1).
7:          Compute $S_{l,c}^{(global)} = S_{l,c}(\theta_{prev}, \mathcal{D}_k)$ using Eq. (1).
8:          Compute pruning mask: $M_{l,c}^{(p)} = \mathbf{1}[S_{l,c}^{(local)} > \tau_p]$.
9:          Compute freezing mask: $M_{l,c}^{(f)} = \mathbf{1}[|S_{l,c}^{(local)} - S_{l,c}^{(global)}| > \tau_f]$.
                                           ▷ Apply Pruning
10:      Set adapted model $\tilde{\theta}^{(k)} = \theta \odot \mathbf{M}^{(p)}$.
                                ▷ Local Fine-tuning with Selective Freezing
11:      **for** epoch $e = 1, \ldots, E_{ft}$ **do**
12:          Fine-tune $\tilde{\theta}^{(k)}$ on dataset $\mathcal{D}_k$.
13:          Freeze channels where $M_{l,c}^{(f)} = 0$ during updates.
14:      Initialize BN layers in $\tilde{\theta}^{(k)}$.
15:      Replace classifier head in $\tilde{\theta}^{(k)}$ if applicable.
16:      Send locally updated $\tilde{\theta}^{(k)}$ to server.
17: Aggregate received models to form final unlearned model $\tilde{\theta}$.
18: **return** $\tilde{\theta}$.

---

## A.1    Detailed Setup

**Datasets and Models.**    We conducted the experiments on **Domain-Digits** with the *CNN* model consisting of 3 convolution layers, 2 max-pool layers and 3 fully connected layers; **Office-Caltech-10** datasets with *VGG16* for unlearning on the training and *ResNet50 & ViT* for unlearning on the fine-tuning; and **PACS** with VGG16. **Domain-Digits** Hull (1994); LeCun et al. (1998); Netzer et al. (2011); Roy et al. (2018); Ganin & Lempitsky (2015) has 5-domain 10-class digital classification image dataset, including {MNIST (M), SVHN (SV), USPS (U), SynthDigits (SD), MNIST-M (MM)}, containing both single-channel and three-channel data simultaneously. **Office-Caltech-10** Gong et al. (2012) has 4-domain 10-class three-channel goods classification image dataset, including {Amazon (A), Caltech (C), DSLR (D), Webcam (W)}. **PACS** Li et al. (2017) has 4-domain 7-class three-channel classification image dataset, including {Art Painting (AP), Cartoon (CT), Photo (PT), and Sketch (SK)}. For all datasets, we set the ratio of the training set to the test set at 8:2. The network structure Settings are detailed in Appendix.

**Baselines.** In additional to Original Learning Model (OL) as our benchmark, we primarily examined two unlearning approaches: precise unlearning requiring model retraining and approximate unlearning based on fine-tuning the existing model. Precise unlearning methods include: (1) Retrain (RE): the gold-standard baseline for complete data removal; (2) Rapid Retrain (RR) Liu et al. (2022): a strategy to fully eliminate target samples from a well-trained global model by leveraging an approximate loss function; (3) FedEraser (FE) Liu et al. (2021): an efficient framework for erasing client-specific data impacts in FL through historical parameter update analysis. Approximate unlearning methods encompass: (4) Increase Loss (IL) Halimi et al. (2022): a federated unlearning technique employing reverse training at the forgetting client by maximizing local empirical loss through inverted learning dynamics; (5) Class-Discriminative Pruning (CP) Wang et al. (2022): a CNN channel pruning mechanism guided by TF-IDF scoring to selectively remove network channels while minimizing information loss during federated unlearning; (6) FedSalUn (FS) Fan et al. (2023): our adapted approach modifying SalUn (originally a centralized unlearning method) for FL scenarios, categorized as an approximate unlearning solution. SalUn shuffle the labels in dataset $\mathcal{D}_f$, generates a saliency mask by leveraging bit-wise weights of model update differences, and subsequently adjusts the update weights to obtain the unlearned model.

**FL settings.** During the FL process, for each dataset, we assign an entire domain of data to each client. The local update epoch is set to 10, and the conmmunication rounds are the same as unlearning rounds. We set unlearning rounds as 10 for Domain-Digits and 40 for Office-Caltech-10. We use the cross-entropy loss function and an SGD optimizer with a learning rate of 0.01 for local updates.

## A.2 EXTENSIVE EXPERIMENTS

Findings in Table 3 across datasets are as follows: (i) *Domain-Digits, forget SynthDigits*: `FU-DWS` achieves the best Overall, improves both Retain metrics and Test accuracy, and reduces BKD. (ii) *Domain-Digits, forget MNIST-M*: `FU-DWS` again ranks first on Overall with concurrent Retain and Test gains while keeping Forget controlled. (iii) *Office-Caltech-10, forget Webcam*: `FU-DWS` delivers the highest Overall, consistent positive shifts on Retain and Test, and a lower BKD. (iv) *PACS, forget Cartoon*: `FU-DWS` retains the top Overall, increases Retain and Test, and maintains competitive Forget.

Across datasets and forgotten domains, `FU-DWS` consistently attains the best Overall score while jointly improving retention and generalization and suppressing residual signals on the forgotten domain. The results support that activation-aware surgery is a reliable and transferable mechanism for federated domain unlearning.

## A.3 LAYER VS. CHANNEL VS. WEIGHT

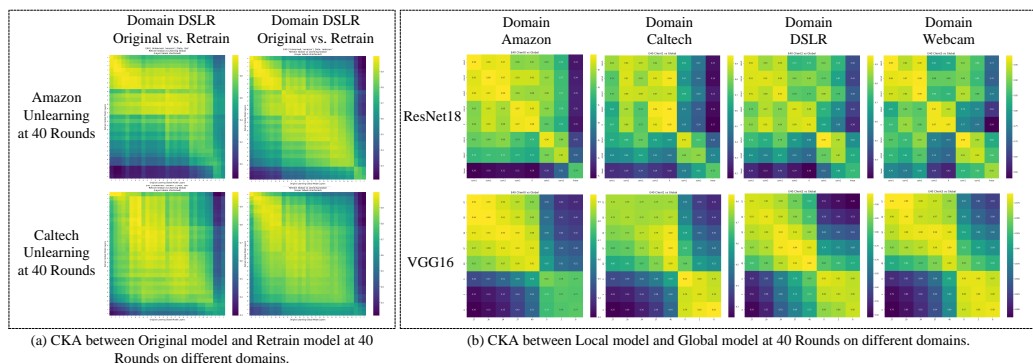

(a) CKA between Original model and Retrain model at 40 Rounds on different domains.

(b) CKA between Local model and Global model at 40 Rounds on different domains.

Figure 6: CKA comparison to understand why not layer-level weight surgery.

To realize effectively federated domain unlearning, an intuitive strategy for efficient federated domain unlearning, without incurring substantial communication overhead (e.g., from transmitting data prototypes), involves mining domain-aware information directly from the models. To the best of our knowledge, no definitive theory currently directly links specific parameter importance metrics to the broader concept of "domainness" in a federated setting. However, extensive empirical studies

Table 3: Average performance (%) on last 5 epochs of various datasets.

| Methods | Retain Performance | | Forget Performance | | Test Performance | | | Overall (↑) |
|---|---|---|---|---|---|---|---|---|
| | $Acc_S(\uparrow)$ | $Acc_U(\uparrow)$ | $Acc_F(\downarrow)$ | $BKD(\downarrow)$ | $Acc_S(\uparrow)$ | $Acc_U(\uparrow)$ | $Acc_F(\uparrow)$ | Performance |
| Domain-Digits & Unlearning Domain = SynthDigits | | | | | | | | |
| RE | 85.18 | 97.95 | 44.3 | 0.57 | 72.20 | 95.43 | 81.64 | - |
| RR Liu et al. (2022) | -25.95 | -0.81 | -5.21 | +2.22 | -23.04 | -0.84 | -11.35 | -2.66 |
| FE Liu et al. (2021) | -4.31 | -0.72 | -0.45 | +0.32 | -0.91 | -0.04 | -1.43 | -0.56 |
| IL Halimi et al. (2022) | +8.85 | +1.87 | +4.38 | -0.34 | +5.73 | +2.48 | +5.79 | +0.59 |
| CP Wang et al. (2022) | +6.05 | +1.38 | +3.04 | +0.43 | +4.13 | +1.61 | +3.81 | +0.30 |
| FS Fan et al. (2023) | +8.82 | +1.67 | +23.98 | +40.27 | +5.86 | +2.02 | +6.58 | -2.50 |
| Ours | +10.14 | +1.82 | +2.92 | -0.22 | +6.56 | +2.51 | +4.68 | **+0.72** |
| Domain-Digits & Unlearning Domain = MNIST-M | | | | | | | | |
| RE | 91.02 | 99.42 | 38.78 | 2.46 | 79.67 | 99.35 | 69.59 | - |
| RR Liu et al. (2022) | -22.98 | -0.76 | -7.82 | +1.51 | -20.95 | -0.76 | -15.57 | -2.69 |
| FE Liu et al. (2021) | +0.38 | -0.88 | +0.35 | +0.69 | +1.45 | -0.57 | +0.16 | -0.17 |
| IL Halimi et al. (2022) | +5.94 | +0.50 | +8.31 | -1.00 | +3.07 | +1.19 | +11.19 | +0.45 |
| CP Wang et al. (2022) | +5.00 | +0.33 | +3.33 | +0.74 | +2.68 | +0.84 | +3.67 | +0.24 |
| FS Fan et al. (2023) | -2.98 | -1.18 | +24.87 | +23.90 | -2.96 | -0.97 | +3.67 | -3.35 |
| Ours | +7.30 | +0.54 | +1.66 | +0.02 | +5.12 | +0.93 | +2.52 | **+0.60** |
| Office-Caltech-10 & Unlearning Domain = Webcam | | | | | | | | |
| RE | 84.65 | 91.74 | 42.03 | 2.26 | 80.62 | 84.44 | 79.65 | - |
| RR Liu et al. (2022) | -37.75 | -54.55 | -11.39 | +37.36 | -38.13 | -58.89 | -45.52 | -5.57 |
| FE Liu et al. (2021) | -1.46 | -28.26 | -14.09 | -1.51 | -0.31 | -26.66 | -23.79 | -0.94 |
| IL Halimi et al. (2022) | +2.30 | +0.17 | -1.27 | -1.51 | -1.04 | -1.67 | -3.79 | 0.43 |
| CP Wang et al. (2022) | +5.14 | +3.31 | +0.34 | -2.26 | +3.23 | +1.67 | -2.07 | 0.75 |
| FS Fan et al. (2023) | -7.18 | +0.00 | +3.04 | -0.94 | -11.46 | -2.22 | -0.69 | -0.12 |
| Ours | +7.05 | +2.48 | +1.86 | -1.51 | +4.48 | +3.34 | -5.52 | **+0.78** |
| PACS & Unlearning Domain = Cartoon | | | | | | | | |
| RE | 88.88 | 75.50 | 33.40 | 16.52 | 77.15 | 70.50 | 48.66 | - |
| RR Liu et al. (2022) | -27.14 | -40.84 | -15.90 | -1.23 | -15.67 | -15.67 | -29.13 | -1.72 |
| FE Liu et al. (2021) | -76.95 | -57.07 | -26.41 | -16.52 | -65.28 | -45.89 | -34.86 | -3.87 |
| IL Halimi et al. (2022) | +2.17 | +8.40 | +12.09 | +0.85 | +3.09 | +9.40 | +7.09 | +0.53 |
| CP Wang et al. (2022) | +6.21 | +11.19 | +9.69 | +6.24 | +8.55 | +9.87 | +7.31 | **+0.70** |
| FS Fan et al. (2023) | +2.04 | +4.47 | +15.12 | +2.29 | +3.03 | +8.46 | +5.95 | +0.31 |
| Ours | 5.59 | 10.29 | 9.92 | 1.95 | 7.46 | 9.03 | 4.67 | +0.69 |

in cross-domain transfer learning and other domain-specific tasks demonstrate that manipulations of model parameters can indeed capture and reflect spatial information pertinent to certain domains. This motivates our exploration of domain-aware information within model parameters, guided by observations and validated through approximate theoretical reasoning.

To investigate this, we analyzed parameter importance at three granularities: layer-level, channel-level, and weight-level. Figure 7 visually supports our rationale. Parameter importance is often gauged by metrics computed at a specific granularity, followed by ranking or difference comparison. For layer-level analysis, we employed Centered Kernel Alignment (CKA) Kornblith et al. (2019) to measure inter-layer similarity between models. For channel-level, we computed mean channel activations. For weight-level, we performed direct weight differencing.

Figure 7(a) illustrates why layer-level operations alone may be insufficient. We compared the CKA between an original model and a retrained model (gold standard for unlearning) on a specific retain domain using VGG16. The CKA differences were modest, highlighting the subtlety of unlearning. Domain generalization and spatial overlap between domains complicate direct unlearning via coarse layer manipulations.

Figure 7(b) highlights domain heterogeneity in federated scenarios. We compared layer-wise CKA between local models and the global model after the final training round on a specific domain. Two key observations emerged across CNN, VGG16, and ResNet18: **First**, federated domain data induces significant CKA dissimilarity between feature extraction layers and the classifier, aligning with the

cross-domain setting of disparate feature spaces but a common label space. **Second**, notable CKA differences exist between local and global models. This underscores the complexity introduced by domain heterogeneity in FL, but also suggests an opportunity: can we trace domain-specific characteristics through these local-global discrepancies?

From the experimental results, this consideration is valid. Therefore, although we did not directly use all layer-level operations, we still derived an effective domain-aware design based on research at the layer level.

In recent years, many studies have explored directly using model parameters Chen et al. (2024) or gradients Fan et al. (2023) to compute model importance scores, as exemplified by the **SalUn** method discussed in the main text. We also investigated this approach. However, we found that even when masks are randomly chosen, a certain level of federated unlearning can still be achieved. We additionally tested **random sampling masks** and **per-layer masks sampled from specific distributions**, alongside our model-difference-based mask (**FedSalUn**). Under the same mask ratio and the same MNIST domain-unlearning setup, FedSalUn's performance on the Forget side (measured by Forget-set accuracy $Acc_F$ and $BKD$) differed from the random-sampling mask by -0.09 (BKD) and -0.08 ($Acc_F$), and from the per-weight exponential-sampling mask by -0.06 (BKD) and +0.50 ($Acc_F$). These small and partially inconsistent differences do not provide convincing evidence that model-difference meaningfully guides domain unlearning. Consequently, we abandoned the idea of implementing domain unlearning at the weight level.

In conclusion, we believe that activation-based weight surgery is the only approach capable of truly achieving effective unlearning.

### A.4 LAYER OPERATION: RESETTING BATCHNORM AND CLASSIFIER

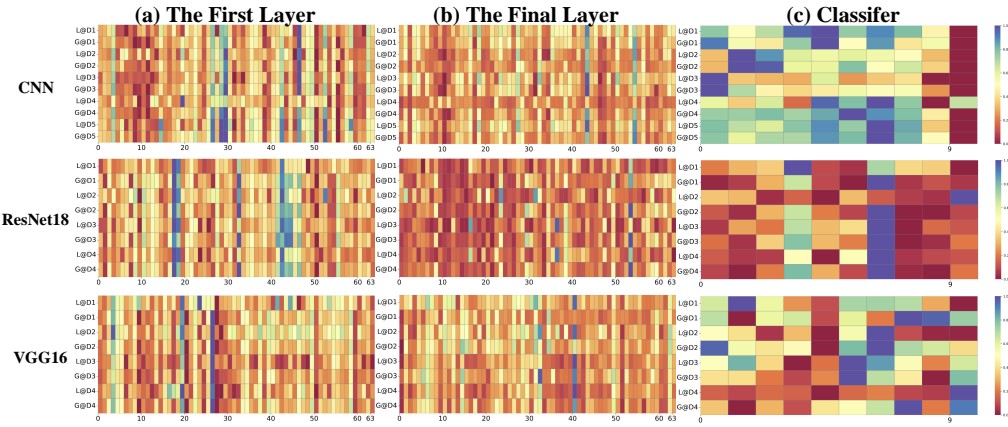

Figure 7: Activation comparison to understand why replace classifier cross various Deep Neural Networks. Vertical comparison can observe the differences between the local and global, as well as among the clients.

In our approach, resetting Batch Normalization (BN) layers and replacing the local classifier head are crucial steps for effective federated domain unlearning. Inspired by Li et al. (2021); Lee et al. (2023), we reset BN layers to mitigate domain shift caused by heterogeneous data distributions across federated clients. Specifically, BN statistics captured during global model training predominantly reflect aggregate, cross-domain distributions. Consequently, utilizing global BN parameters directly can deteriorate the performance on local domains after domain-specific pruning. By reinitializing BN layers, our method allows each retain client to recalibrate normalization statistics based exclusively on local data distributions, thus effectively addressing domain heterogeneity.

Additionally, we observed through activation analysis as Figure 7 and CKA metrics as Figure 6 that classifiers exhibit distinct activation patterns and significant discrepancies from feature extraction layers, especially when subjected to domain shift. Considering domain-specific consistency in label spaces, we argue that replacing the global classifier with a local one further strengthens domain-aware

Table 4: Performance of ablation study (%) from last 5 epochs on Office-Caltech-10.

| Methods | Forget Accuracy ($\downarrow$) | | Retain Accuracy ($\uparrow$) | | | Overall($\uparrow$) |
| --- | --- | --- | --- | --- | --- | --- |
| | Webcam | $BKD$ | Amazon | Caltech | DSLR | Performance |
| Ours | 43.80±1.48 | 0.66±0.60 | 91.63±1.43 | 99.36±0.42 | 94.71±2.22 | **48.25** |
| Ours-No Pruning | 45.36±1.88 | 0.94±1.33 | 83.09±3.00 | 98.56±0.66 | 95.87±0.90 | 47.81 |
| Ours-No Freezing | 42.53±1.15 | 0.66±0.43 | 84.09±3.59 | 96.89±1.54 | 97.02±1.34 | 46.96 |
| Ours-No Resetting | 43.08±1.87 | 1.41±0.97 | 88.71±2.34 | 98.05±0.95 | 96.78±1.67 | 46.24 |

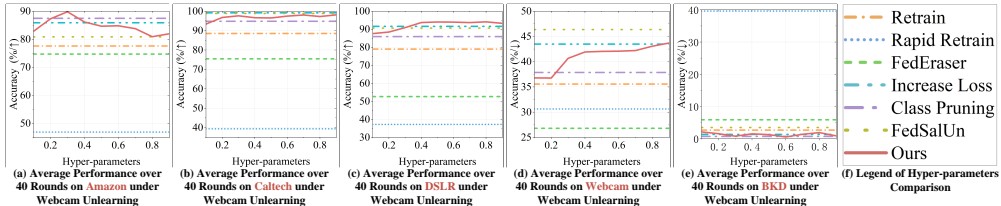

Figure 8: Hyper-parameters performance over 40 rounds of each domain under Webcam unlearning.

specialization. Our ablation study empirically validates these operations, demonstrating marked improvements in both unlearning efficiency and retain-domain performance.

Table 4 confirms the effectiveness of each component in our framework, illustrating clear performance degradation when removing any individual step, demonstrating the necessity of our integrated pruning and freezing strategies.

## A.5  $\tau_p$ & $\tau_f$: A HUGE SEARCH SPACE AND POTENTIAL FOR PERSONALIZED ADJUSTMENT

Our experiments set both pruning ($\tau_p$) and freezing ($\tau_f$) thresholds in increments of 10% within the [0.1, 0.9] range. However, the actual hyperparameter search space for these thresholds is significantly larger.

Figure 8 presents an extensive hyper-parameter analysis, where average performance across all training rounds is depicted to emphasize both high accuracy and rapid recovery. Results indicate robust performance across a wide range of hyper-parameter choices, maintaining consistently high $\text{Acc}_R$ while ensuring low $\text{Acc}_F$ compared to all approximate baselines, further affirming the robustness and reliability of our method.

Despite our experimental setting of equal pruning and freezing thresholds ($\tau_p = \tau_f$), these parameters can be independently optimized. Given our coarse hyper-parameter granularity and large unexplored parameter space, our method demonstrates significant potential and robustness without extensive fine-tuning. Detailed discussions on hyper-parameter sensitivity and potential for personalized adjustment are further provided in the Appendix.

Moreover, we empirically observed performance variations when $\tau_p \neq \tau_f$, suggesting distinct roles for pruning and freezing in our unlearning and recovery framework. Prior studies Lee et al. (2023) highlighted the variability of channel importance across different layers and domains. Hence, identifying domain-channel-layer interactions could enable further refined hyperparameter tuning, enhancing domain-specific feature extraction and improving overall model robustness. This perspective not only underscores the inherent flexibility of our method but also provides a strong foundation for extending `FU-DWS` to personalized federated scenarios.

## A.6  RESULTS UNDER SHARED-DOMAIN SETTING

Despite domain overlap across clients, **FU-DWS** is able to **selectively suppress the influence of the forgetting client**, while preserving performance on retained domains. This demonstrates that:

1. **FU-DWS** remains effective even when **multiple clients share the same domain**;

Table 5: Performance on domain amazon unlearning. Each domain has 2 Clients (8 Clients in Total) and we unlearned on 2 clients of Amazon for whole domain erasing.

| Met. | Retain Performance | | | Forget Performance | | Test Performance | | | | Overall (↑) |
| | $Acc_C$ ↑ | $Acc_D$ ↑ | $Acc_W$ ↑ | $Acc_F$ ↓ | BKD↓ | $Acc_C$ ↑ | $Acc_D$ ↑ | $Acc_W$ ↑ | $Acc_F$ ↑ | Performance |
|---|---|---|---|---|---|---|---|---|---|---|
| RE | 85.20 | 74.10 | 94.69 | 35.07 | 31.69 | 44.62 | 59.44 | 87.59 | 40.42 | - |
| IL | **+7.22** | +6.30 | **+4.47** | +10.86 | +9.88 | **-8.09** | 0.00 | **+6.00** | **-1.46** | -6.24 |
| Ours | 0.00 | **+10.36** | +2.04 | **+8.51** | **+7.15** | -10.22 | **+10.56** | +3.10 | -5.42 | **-5.24** |

2. It can isolate **fine-grained, client-specific domain signatures** through **activation-aware saliency**;

3. It avoids unintentional degradation of **domain-invariant features**, directly addressing the concern about correlated channels.

These results indicate that channel-level intervention in **FU-DWS** is **not based on a naive independence assumption**, but rather guided by **implicit inter-channel correlation captured in the activation space**, which is sufficient to support reliable and fine-grained domain-level unlearning in practice.

## A.7   LLM USAGE

We used an LLM only to polish the language to conform with the ICLR 2026 policies on LLM usage; all scientific content was authored and verified by the human authors.

