# OpenReview forum: "FU-DWS: Effective Federated Domain Unlearning via Domain-aware Weight Surgery"
_ICLR.cc/2026/Conference — Submitted to ICLR 2026_

### Official Review · Reviewer_eKXD · 2025-10-28

**Soundness:** 3
**Presentation:** 3
**Contribution:** 3
**Rating:** 6
**Confidence:** 3

**Summary:**

This paper systematically addresses the novel and critical challenge of Federated Domain Unlearning, which involves removing the influence of an entire data domain from a well-generalized federated learning model. The authors propose FU-DWS, a novel method grounded in the key observation that channel-level activations in a neural network are highly correlated with domain-specific knowledge. FU-DWS first identifies and prunes channels with high sensitivity to the domain targeted for unlearning, and then selectively freezes stable channels crucial for the retained domains during local recovery. Extensive experiments on multiple heterogeneous datasets demonstrate that FU-DWS significantly outperforms six existing baselines by achieving a superior balance among recovery performance on retained domains, effective forgetting of the target domain, and generalization capability on unseen data.

**Strengths:**

1. This work is the first to systematically define and tackle the problem of federated domain unlearning.

2. The proposed method, FU-DWS, enables precise unlearning by accurately identifying and pruning domain-salient weights. Its dual mechanism of pruning and selective freezing effectively removes the target domain's knowledge while ensuring rapid and stable recovery of performance on retained domains.

3. The method demonstrates state-of-the-art effectiveness across multiple heterogeneous datasets and model architectures. It excels in achieving an optimal trade-off between recovery, forgetting, and generalization.

**Weaknesses:**

1. The method's performance depends on the selection of pruning and freezing thresholds. Although the approach shows some robustness, identifying the optimal values requires tuning. The paper's mention of a "huge search space" suggests that this could impose a non-trivial tuning burden in practical deployments.

2. The evaluation of unlearning effectiveness relies on indirect metrics like accuracy on the forgotten domain and backdoor attack success rate. These methods lack a direct, statistical guarantee that the domain's information has been completely and verifiably erased from the model's parameters.

3. The paper does not sufficiently analyze scenarios where multiple domains have significant feature space overlap. In such cases, the proposed pruning strategy might lead to substantial collateral damage and performance loss on retained domains.

4. The method assumes that all clients within a single domain share an identical underlying data distribution. This simplification may overestimate the method's efficacy in real-world cross-silo settings where data distribution can vary even among clients from the same organization.

5. The requirement for clients to compute and compare activation statistics, while not involving raw data, could potentially leak information about domain characteristics. Furthermore, the reliability of these activation-based metrics is questionable for domains with very few local samples.

**Questions:**

1. How can the pruning and freezing thresholds be adaptively and automatically determined based on model architecture and data characteristics to enhance practicality?

2. What statistical tests or verification frameworks can be developed to provide provable guarantees of complete domain-level forgetting?

3. What is the true computational and memory overhead of calculating activation statistics on resource-constrained edge devices, and how can this process be optimized?

4. Eq.3 introduces the pruning score without further explanation or motivation. This formula is critical for understanding the pruning step, could you clarify how it is derived and why it’s defined that way?

---

> ### Author Response · Authors · 2025-11-28
> **Response to Reviewer eKXD (1/4)**
>
> > ResponseQ1: Adaptive and automatic thresholds (($\tau_p$, $\tau_f$))
>
> We thank the reviewer for this insightful suggestion and fully agree that **adaptive, data-driven thresholds** are an important direction to further improve practicality.
>
> In the current version, FU-DWS uses **activation-based saliency distributions** to set the pruning and freezing thresholds ($\tau_p$) and ($\tau_f$). In the revision, we further analyzed the **hyperparameter sensitivity** and show that FU-DWS is **robust to a broad range of percentile-based thresholds**, which naturally suggests an adaptive scheme:
>
> - We compute the saliency scores ($S_{l,c}$($\theta$, $D_k$)) (Eq. 3) for each layer and channel,
> - Apply layer-wise min–max normalization to obtain scores in ([0, 1]),
> - Then choose ($\tau_p,\tau_f$) as **percentiles of the empirical score distribution** (e.g., prune bottom ($p\%$), freeze top ($q\%$)).
>
> As shown in our sensitivity experiments (Figure 5 and additional results in Appendices 3 & 4), varying ((p, q)) across a wide range yields **stable unlearning and recovery performance**, indicating that FU-DWS is **not highly sensitive to the exact threshold values**.
>
> This makes it natural to design an **automatic thresholding strategy** based on:
>
> - **Score statistics**: choosing (p, q) from simple rules on the score histogram (e.g., Otsu-style separation, or fixed quantiles such as 20–30% for pruning and 60–80% for freezing);
> - **Model/depth awareness**: using slightly higher pruning percentiles in deeper layers where redundancy is typically higher;
> - **Optional validation-based refinement**: when a small validation split from retained domains is available, one can select ((p, q)) that best balances “forgetting strength vs. retain accuracy” along the Pareto front.
>
> We acknowledge that we have not fully implemented a *completely automatic* selection pipeline in this submission. We therefore (i) clarify in the paper that FU-DWS already shows **low sensitivity** under percentile-based choices, and (ii) explicitly list **adaptive threshold learning** as an important direction for future work (e.g., validation-driven or Bayesian selection), which is fully compatible with our current design.

---

> ### Author Response · Authors · 2025-11-28
> **Response to Reviewer eKXD (2/4)**
>
> > ResponseQ2: Verification and guarantees of domain-level forgetting
>
> We very much appreciate this question and agree that **formal verification for domain-level unlearning** is a highly important and challenging direction.
>
> Our current work focuses on **practical, activation-aware unlearning** with strong empirical evidence, but **does not claim formal guarantees of complete domain-level forgetting**. We have clarified this limitation in the revision.
>
> We are aware of a recent ICLR paper (ref1) that also considered **Federated Domain Unlearning (FDU)** and attempted to design verification methodologies. However, as we noted, this work was eventually withdrawn and did not convince reviewers, which further underscores how **non-trivial** it is to rigorously verify domain-level forgetting, especially under the setting we consider:
>
> > *label spaces can overlap across domains while feature spaces differ significantly*,
> >  which makes it fundamentally harder to define and certify “complete forgetting” at the domain level.
>
> We believe our findings and formulation can serve as a **starting point** for future verification-oriented research:
>
> - Our FDU setting explicitly characterizes the case where **different domains share labels but differ in feature distributions**, which is more realistic but also more difficult to verify.
> - Our activation-based saliency analysis reveals which channels respond differently across domains/clients, which could be used as a **feature-level attack surface** for future statistical tests (e.g., domain membership inference, feature attribution tests over domain distributions).
> - One potential direction is to design **hypothesis tests in the feature space**: given a candidate “forgotten” domain, one can test whether any classifier built on intermediate features can still reliably distinguish this domain from the retained ones. Another direction is to explore **influence-function–based certificates**, bounding the residual effect of the forgotten domain on the loss over retained distributions.
>
> We explicitly add a discussion in the “Limitations and Future Work” section that:
>  (i) FU-DWS currently provides **empirical forgetting**, not formal certificates;
>  (ii) **feature-space adversarial testing and statistical verification for FDU** is an open and valuable research direction, to which our FDU formulation and observations provide concrete challenges and design hints.
>
> [ref1] Towards Federated Domain Unlearning: Verification Methodologies and Challenges

---

> ### Author Response · Authors · 2025-11-28
> **Response to Reviewer eKXD (3/4)**
>
> > Response Q3:  Computational & memory overhead of activation statistics on edge devices
>
> We thank the reviewer for raising this practical concern.
>
> **Computation.**
>  In FU-DWS, activation statistics ($S_{l,c}$) are obtained by:
>
> - running **forward passes only** (no extra backprop), and
> - registering lightweight hooks on Conv/Linear layers to accumulate per-channel mean absolute activations.
>
> Thus, the **computational overhead is essentially one extra forward sweep per calibration round**, which is small compared to local training epochs. In practice, we reuse the same forward passes used for local training or evaluation, so that the additional cost is limited to simple accumulation and averaging operations over channel outputs.
>
> **Memory.**
>  The memory overhead for storing activation statistics is:
>
> - ($\mathcal{O}(\sum_l C_l)$) scalars, where ($C_l$) is the number of channels/neurons in layer (l),
> - which is typically **much smaller than** the memory needed for model weights and feature maps.
>
> We only keep **aggregated scalars per channel**, not full activation maps. Compared to dense FU baselines like **Increase Loss (IL)** (ref3), which must keep multiple full model copies and dense gradients in memory, FU-DWS adds only a **tiny vector of per-channel statistics** and two bit-masks.
>
> In terms of overall memory and computation, we have already shown (see our IL vs FU-DWS comparison) that:
>
> - IL requires ($\approx (2-3)w$) parameters in memory and ($O(w)$) per-step operations;
> - FU-DWS requires (\approx (2!-!3)(1-\rho)w) parameters and ($O((1-\rho)w)$) operations, where ($w4$) is the total parameter count and (\rho) the pruning ratio,
>
> and we further demonstrated a **~1.5× memory reduction** on ResNet-50 with ($\rho=0.3$), with larger gains on bigger backbones.
>
> **Optimization on edge devices.**
>  To further optimize the collection of activation statistics on resource-constrained devices, we discuss in the revision that one can:
>
> - **Sample a subset of layers** (e.g., last few blocks or key bottleneck layers) instead of all layers.
> - Use **small calibration batches** and a limited number of steps (e.g., a few mini-batches), as we only need approximate statistics.
> - Use **mixed precision** or low-precision accumulators for saliency scores (e.g., FP16), as exact high precision is unnecessary.
> - Optionally offload part of calibration (e.g., for heavier models) to more capable nodes if available in cross-silo settings.
>
> These design choices, together with the inherently low-dimensional nature of the statistics, keep the activation-collection overhead modest even on edge devices.
>
> [ref3] Federated unlearning: How to efficiently erase a client in fl?

---

> ### Author Response · Authors · 2025-11-28
> **Response to Reviewer eKXD (4/4)**
>
> > ResponseQ4: Clarification and motivation of Eq. 3 (pruning score)
>
> We appreciate the request for clarification. In the revision, we have added a formal and detailed description of the activation-based saliency metric. For layer (l) and channel (c) on client (k), we define:
>
> $$
> S_{l,c}(\theta, D_k)
> = \frac{1}{|D_k|} \sum_{x \in D_k} mean_{h,w} \left| A*{l,c}(x; \theta) \right|
> $$
>
>
> where ($A_{l,c}(x; \theta)$) denotes the activation map produced by channel (c) under input (x), and spatial averaging is replaced by feature-dimension averaging for linear layers.
>
> The design is motivated by two principles:
>
> 1. **Activation magnitude as importance proxy.**
>     Prior work in pruning and quantization (e.g., activation-aware quantization such as AWQ) demonstrates that **channels with larger average activation magnitudes contribute more to representing the current data distribution**, while low-activation channels are often redundant or domain-specific. By averaging over samples in (D_k), (S_{l,c}) estimates the **expected contribution** of channel (c) to client (k)’s domain.
> 2. **Stability and robustness.**
>     We use **absolute values** to avoid cancellation between positive and negative responses and perform **layer-wise min–max normalization** so that saliency scores lie in ([0, 1]) per layer. This allows us to compare channels within the same layer in a scale-invariant way and yields robust, interpretable thresholds for pruning ($\tau_p$) and freezing ($\tau_f$).
>
> We emphasize that FU-DWS does *not* require this score to perfectly separate domains. Instead, it serves as a **soft, activation-aware indicator** that guides:
>
> - **Pruning:** removing channels with high saliency for the forget domain relative to the retained domains;
> - **Freezing:** preserving channels that show stable saliency across global/local models, which likely correspond to domain-invariant features and help recovery.
>
> Combined with the recovery stage, Eq. 3 provides a simple yet effective mechanism to “steer” unlearning toward domain-sensitive features while maintaining domain-invariant capacity.

---

### Official Review · Reviewer_8vgk · 2025-10-30

**Soundness:** 2
**Presentation:** 2
**Contribution:** 2
**Rating:** 4
**Confidence:** 5

**Summary:**

This paper introduces FU-DWS, an algorithm for federated domain unlearning that targets domain-level knowledge removal in cross-silo federated learning. Through a two-stage "weight surgery" (pruning and selective freezing of activation-sensitive channels), the method aims to efficiently remove knowledge tied to one domain while preserving generalization and utility on the remaining domains. Extensive experiments on three multi-domain datasets and six baselines demonstrate that FU-DWS achieves competitive forgetting, strong recovery on retained domains.

**Strengths:**

1. The paper is well organized and clearly written overall, with quantitative analyses in figures and tables that effectively support the main arguments.
2. The motivation targets a realistic and practically significant problem in federated learning, particularly the challenge of domain-level unlearning under privacy and regulatory constraints.
3. The proposed approach is conceptually straightforward and practically feasible, making it easy to implement and integrate into existing federated learning systems.

**Weaknesses:**

1.  The approach heavily relies on activation magnitudes to estimate domain sensitivity and uses it as the sole signal. This reliance may be unstable under different network architectures, normalization layers, or training dynamics. Furthermore, if activation differences alone are insufficient to fully characterize this distinction (e.g., some domain-specific knowledge may not manifest as significant activation differences, or some shared knowledge might occasionally show large activation differences), the effectiveness of this method would be significantly compromised.
2.  In scenarios where domain knowledge overlaps, these domains may not be distinguishable solely by the activation magnitudes of different channels. The authors need to further discuss whether the channel pruning strategy might inadvertently damage knowledge belonging to retained domains when the forget domain and retained domains have similar knowledge.
3.  Some hyperparameters in this paper (e.g., $\tau_p$, $\tau_f$) rely on manual grid search, which limits their applicability in highly complex scenarios. Additionally, the related sensitivity experiments were only validated on a small scale using the Webcam dataset (Figure 8, Appendix A5).
4.  Most of the methods compared in this paper are not specifically focused on domain unlearning and were primarily proposed before 2023. Thus, they may not represent the state-of-the-art level, making the comparison potentially unfair.
5.  The experiments are primarily conducted on relatively small and image-based datasets (Domain-Digits, Office-Caltech-10, PACS). It remains unclear whether the method scales to large-scale, high-dimensional, or non-vision tasks, such as text or tabular federated learning scenarios.

**Questions:**

1. Can the authors clarify the mathematical justification and potential pitfalls of using mean-absolute activations ($S_{l,c}$) as a proxy for domain saliency? Given the observed overlap or instability of activation distribution across domains, how robust is this metric to adversarial domain mixing/noise?
2. Have you tested FU-DWS on settings with partial domain overlaps, or where the forget domain is semantically strongly related to the remaining ones? If not, can you comment on expected performance?

---

> ### Author Response · Authors · 2025-11-28
> **Response to Reviewer 8vgk (1/3)**
>
> >  Response to Q1: Mathematical justification of ($S_{l,c}$) as a domain-saliency proxy.
>
> In the revision, we have added a formal and detailed description of the activation-based saliency metric. For layer (l) and channel (c) on client (k), we define:
>
> $$
> S_{l,c}(\theta, D_k) = \frac{1}{|D_k|} \sum_{x \in D_k}Mean_{h,w} \left| A*{l,c}(x; \theta) \right|
> $$
>
> where ($A_{l,c}(x; \theta)$) denotes the activation map produced by channel (c) under input (x), and spatial averaging is replaced by feature-dimension averaging for linear layers.
>
> This formulation is motivated by a well-established insight in representation learning and quantization literature (e.g., AWQ): **activation magnitude reflects the degree to which a channel participates in representing the underlying data distribution**. Thus, a channel that consistently produces high activations on domain ($D_k$) is more likely to encode **domain-correlated features**. The use of mean-absolute activations further improves stability by avoiding positive/negative cancellation.
>
> In practice, we register forward hooks on all Conv2d/Linear layers and accumulate per-channel mean activation values across batches, followed by layer-wise min–max normalization. This ensures scale-invariance and allows meaningful comparison within each layer.
>
>
> > Response to Q1/W1: Robustness to activation overlap, noise, and adversarial domain mixing.
>
> We emphasize that FU-DWS does **not** rely on ($S_{l,c}$) being perfectly separable across domains. Our method is explicitly designed as:
>
> * a **soft, statistical indicator**,
> * not a hard classifier of domain ownership.
>
> Even if multiple domains activate overlapping channels, it is the **relative divergence between the local model and the global model** that matters. Channels whose activation profiles significantly differ between these two views are flagged as domain-sensitive. This comparison is intrinsically robust to moderate overlap and noise, because it measures *contextual inconsistency* rather than absolute magnitude alone. Furthermore, we perform **layer-wise normalization**, which reduces the impact of extreme values and noisy outliers, and ensures that channels are compared relative to their layer-wise activation range.
>
> At last, we have built-in safety net: unlearning + recovery as a closed-loop mechanism. Importantly, FU-DWS **does not require ($S_{l,c}$) to be perfectly precise**. Our framework is a **closed-loop system** consisting of:
>
> * **Pruning stage:** removes channels likely associated with the forget domain
> * **Freezing stage:** protects activation-stable, domain-invariant channels
> * **Recovery stage:** quickly restores performance via retained-domain clients
>
> Even if a small number of channels are imperfectly classified due to overlap or noise, the **subsequent recovery and freezing mechanisms compensate** for such errors. Empirically, this is validated in our experiments, where FU-DWS consistently achieves both effective forgetting and rapid recovery, even in partially overlapping domain scenarios (e.g., shared-domain clients in Office-Caltech-10).
>
> This design trades strict theoretical separation for a **practical, privacy-preserving and robust** solution to domain-level unlearning, without requiring access to the forgotten domain.

---

> ### Author Response · Authors · 2025-11-28
> **Response to Reviewer 8vgk (2/3)**
>
> > Response to Q2/W2: unlearning effectiveness in domain knowledge overlaps
>
> We clarify that FU-DWS remains effective when multiple clients share the same domain, which we validate experimentally. Specifically, in Office-Caltech-10, each domain is split across two clients, forming 8 clients total. We simulate a realistic setting by issuing an unlearning request on only one client holding the target domain, while the other continues to participate. We compared the performance with Top-2 method IL. The result shows as follows:
>
> |Methods|Acc_C↑|Acc_D↑|Acc_W↑|Acc_F↓|BKD↓|Acc_C↑|Acc_D↑|Acc_W↑|Acc_F↑|Overall↑ (without scaling)|
> |:-|:-:|:-:|:-:|:-:|:-:|:-:|:-:|:-:|:-:|:-:|
> |**Unlearning Domain = Amazon; Each domain has 2 Clients (8 Clients in Total)**|||||||||||
> |RE|85.20|74.10|94.69|35.07|31.69|44.62|59.44|87.59|40.42|-|
> |IL|**+7.22**|+6.30|**+4.47**|+10.86|+9.88|**−8.09**|0.00|**+6.00**|**−1.46**|−6.24|
> |Ours|0.00|**+10.36**|+2.04|**+8.51**|**+7.15**|−10.22|**+10.56**|+3.10|−5.42|**−5.24**|
>
> - Despite sharing the same domain, each client’s local training induces unique activation patterns, due to stochastic optimization and local data variations. FU-DWS detects these subtle differences by comparing activations between the local and global models, allowing it to isolate domain-sensitive units specific to the forgetting client, even in partially shared settings.
>
> - Our method does not rely on global domain labels—instead, it prunes weights tied to the forgetting client’s activation profile, while preserving the generalizable parts of the model through freezing. As shown in Table above, this leads to strong forgetting with minimal impact on retained clients, outperforming existing baselines. **This result highlights that FU-DWS supports fine-grained, domain-level unlearning even under domain overlap by activation-aware saliency, which is common in practice.**

---

> ### Author Response · Authors · 2025-11-28
> **Response to Reviewer 8vgk (3/3)**
>
> > Response to W3: hyperparameters in complex scenarios.
>
> Thank you for your valuable feedback. We agree that the threshold parameters ( $\tau_p$ ) (pruning) and ( $\tau_f$ ) (freezing) are important in FU-DWS, and their tuning could affect performance if not handled carefully.
>
> To reduce reliance on manual grid search, we explored a **percentile-based adaptive strategy**, which sets thresholds based on the relative distribution of the activation-aware saliency scores (e.g., pruning the bottom *p%* and freezing the top *q%* channels). As shown in **Figure 5** and additional results across multiple datasets and backbones in **Appendices 3 & 4**, FU-DWS maintains **stable performance across a wide range of percentile settings**, indicating that the method is **not highly sensitive to precise values of ( $\tau_p$ ) and ( $\tau_f$ )**.
>
> While the initial sensitivity visualization was conducted on the Webcam split (Figure 8, Appendix A5), the supplementary experiments confirm that this robustness holds across other domains and datasets as well. We have clarified this in the revision to avoid the impression of limited validation.
>
> We also recognize that different users may prefer different trade-offs between **unlearning strength** and **recovery speed**. We will therefore include a practical guideline in the revised manuscript:
>
> - A **lower ( $\tau_p$**) (i.e., pruning fewer channels) leads to **weaker unlearning but smaller performance drop**.
> - A **lower ( $\tau_f$ )** (i.e., freezing fewer channels) provides **more flexibility during recovery**, but may **slow convergence**.
>
> These adjustments are optional, as FU-DWS already performs robustly under default percentile-based settings across various tasks.
>
> Finally, we agree that automatic threshold tuning is an important future direction. Incorporating adaptive mechanisms (e.g., validation-driven or Bayesian selection) is part of our planned extensions.
>
>
>
> > Response to W4: Most of the methods compared in this paper are not specifically focused on domain unlearning and were primarily proposed before 2023. Thus, they may not represent the state-of-the-art level, making the comparison potentially unfair.*
>
> We thank the reviewer for raising this important concern. Currently, **federated domain unlearning (FDU)** remains an underexplored topic in the literature, and most existing unlearning methods focus on **sample-, class-, or client-level deletion under standard FL settings**, rather than **domain-partitioned clients with shared label spaces**. To our best knowledge, our work is the **first to explicitly formulate and study the FDU problem**.
>
> For this reason, we intentionally included a **diverse set of representative baselines** from federated unlearning and machine unlearning literature (e.g., retraining-based, regularization-based, pruning-based, and feature-based methods), in order to provide a **comprehensive and fair comparison** under our newly-defined setting. Although some of these methods were proposed before 2023, they remain widely-used benchmark approaches in unlearning research.
>
> We agree that continuously updating baselines is important. If accepted, we will actively extend our evaluation to include **newer or closely-related methods**, especially those exploring unlearning in non-IID or domain-shifted scenarios, in the camera-ready version and future extensions.
>
>
> > **Response to W5** *The experiments are primarily conducted on relatively small and image-based datasets (Domain-Digits, Office-Caltech-10, PACS). It remains unclear whether the method scales to large-scale, high-dimensional, or non-vision tasks, such as text or tabular federated learning scenarios.*
>
> We agree with the reviewer that our current experimental evaluation focuses on vision-based and moderate-scale datasets. While these datasets are standard for domain generalization and domain adaptation research, we acknowledge that they do not fully cover **large-scale, high-dimensional, or non-vision tasks**.
>
> The proposed FU-DWS framework, however, is **not tied to vision-specific architecture**. It operates at the level of **neurons/channels and activations in generic neural networks** (Conv and Linear layers), and is therefore applicable to **non-vision architectures, including MLPs and Transformers** commonly used in NLP and tabular learning.
>
> Due to computational and time constraints, we were unable to included large-scale NLP or tabular datasets in the current submission. Nevertheless, extending FU-DWS to **text-based FL (e.g., sentiment analysis, question answering)** and **tabular financial/medical datasets** is a natural and important next step. We will explicitly highlight this in the **“Limitations & Future Directions”** section and plan to expand our experimental scope accordingly in future revisions.

---

### Official Review · Reviewer_JPNY · 2025-10-30

**Soundness:** 3
**Presentation:** 2
**Contribution:** 3
**Rating:** 6
**Confidence:** 3

**Summary:**

This paper proposes a new method for **domain unlearning** in a federated learning setting. The main idea behind this problem is that, upon a request from clients holding data from a given domain, the federated model being learned must behave as it had never seen data from that domain. The main motivation for authors method is detecting parameters that are related to the specific domain. These parameters are detected by comparing the weights of the global model, and local model, which seems reasonable.

**Strengths:**

Overall the method is well motivated, simple and reasonable. I think this is an interesting direction for domain unlearning. The federated setting is also quite appealing and practical.

**Weaknesses:**

While I myself could not find any technical problems with this paper, I think it has many writing problems.

__Problem 1.__ Overall I think Figure 2 is quite difficult to interpret. For instance, Fig 2. (a) shows a heatmap of model activations, but the actual difference and the mechanism for the selection of parameters to be pruned could me made more explicit. The same holds for Fig 2 (b). Furthermore, there is no colorbar indicating what the colors in the heatmap mean.

__Problem 2.__ The function $S\_{l,c}(\theta^{k}, \mathcal{D}\_{k})$ is never explicitly defined in mathematical terms.

__Problem 3.__ Table 1 in the experiments is never mentioned in the text. I suspect the authors made a typo in the last paragraph of Page 7, where they mention Table 3. I think it should be Table 1.

**Questions:**

N/A

---

> ### Author Response · Authors · 2025-11-28
> **Response to Reviewer JPNY**
>
> We thank the reviewer for carefully reading the paper and providing concrete and actionable suggestions. We have revised the manuscript accordingly, as detailed below.
>
> > Response to Problem 1 (Figure 2 is difficult to interpret)
>
> We agree that the original Figure 2 could be difficult to interpret and did not sufficiently explain the mechanism of parameter selection.
>
> In the revision, we have **fully updated Figure 2 and presented it as Figure 1**, with the following improvements:
>
> 1. **Added a colorbar** to the heatmap so that the meaning of activation intensity is explicit.
> 2. **Clarified the visual semantics**: we now clearly indicate which colors correspond to higher activation magnitude and which correspond to lower values.
> 3. **Annotated the pruning and freezing regions more explicitly**, showing how domain-sensitive and domain-invariant parameters are separated.
> 4. **Reorganized the accompanying explanation** into four structured parts:
>     **(i) Algorithm Aiming**,
>     **(ii) Intuition**,
>     **(iii) Observation**, and
>     **(iv) Empirical Study**,
>     in order to guide the reader step by step from motivation to implementation.
>
> These changes significantly improve the readability and interpretability of the figure.
>
>
> > Response to Problem 2 (Function is not defined mathematically)
>
> We agree that the original manuscript did not explicitly define the activation-based saliency function in mathematical form.
>
> In the revision, we have added a formal and detailed description of the activation-based saliency metric. For layer (l) and channel (c) on client (k), we define:
> $$
> S_{l,c}(\theta, D_k)
> = \frac{1}{|D_k|} \sum_{x \in D_k} \text{mean}*{h,w} \left| A*{l,c}(x; \theta) \right|
> $$
>
>
> where ($A_{l,c}(x; \theta)$) denotes the activation map produced by channel (c) under input (x), and spatial averaging is replaced by feature-dimension averaging for linear layers.
>
> We also added **implementation details** explaining that we register forward hooks on Conv2D and Linear layers to collect mean absolute activations across batches, followed by layer-wise min–max normalization. This change improves **clarity, reproducibility, and mathematical completeness**.
>
> > Response to Problem 3 (Table 1 is not mentioned in the text)
>
> We apologize for the incorrect table label reference. In the latest version of the paper, we have completed the correction.

---

### Official Review · Reviewer_iLq4 · 2025-10-31

**Soundness:** 3
**Presentation:** 3
**Contribution:** 3
**Rating:** 4
**Confidence:** 3

**Summary:**

This submission proposes FU-DWS (Federated Domain Unlearning via Domain-aware Weight Surgery), a method designed to selectively remove domain-specific knowledge from federated models while preserving performance on retained domains. The problem addressed is federated domain unlearning, where an entire domain (e.g., hospital or dataset) must be forgotten due to legal or ethical requirements under heterogeneous, cross-silo FL settings. FU-DWS identifies activation-aware domain-salient weights that are highly sensitive to the target domain and performs a two-stage "surgical" operation: pruning domain-specific channels and freezing domain-invariant ones for efficient recovery. Experiments on Domain-Digits, Office-Caltech-10, and PACS show that FU-DWS achieves superior performance over six baselines (Retrain, Rapid Retrain, FedEraser, Increase Loss, Class-Pruning, and FedSalUn), improving unlearning efficiency up to 25× while maintaining higher accuracy on retained domains.

**Strengths:**

++ The activation-based channel saliency approach is novel and intuitively appealing, providing a clear mechanistic interpretation of selective forgetting.

++ The experiments are comprehensive, including multiple datasets, six baselines, and ablation studies verifying each module (pruning, freezing, resetting).

++ FU-DWS achieves state-of-the-art trade-offs among forgetting effectiveness, retained-domain recovery, and efficiency, with consistent performance across CNN and ViT backbones.

**Weaknesses:**

-- The motivation for focusing on federated domain unlearning (FDU) is not well-justified, as the discussion on the limitations of conventional federated unlearning (FU) is not convincing. See C1.

-- The discussion on the unique challenges of FDU compared to conventional FU is not well-justified. See C2.

-- The discussion on related work is not comprehensive and lacks details. See C3.

-- The method design has clear limitations that need to be addressed, but without solutions or discussions. See C4.

Writing Issues (Only a few typos are listed here. Please carefully revise the submission):





Line 17: “Data domains” typically refer to data samples. This sentence needs revision.



Line 19: “Catastrophic forgetting” is a specific term used in continual learning, which refers to the backward transfer (e.g., performance degradation on downstream tasks) across tasks. Please carefully ensure the correctness when using such professional terminology.



Line 62: “existing methods” → lack of references.



Line 63: What is “collateral forgetting?” There is no such term in the cited survey.



Line 83: “Recovery & Forget & Generalization” → “recovery, forgetting, and generalization”.



Line 93: “Retrain” → “retrain”.



Line 129: “1.Retrain Unlearning” → “1. Retrain unlearning”.



Line 132: “. 2. Reverse” → “; 2. Reverse”.

**Questions:**

C1:

- Does “homogenous setting” refer to “single-domain?” Does “heterogeneous setting” refer to “multi-domain?”
- Assume the above statement is true. Then, “single-domain” means all clients have the same data distribution, while “multi-domains” means all clients have different data distributions. The submission claims that the motivation to focus on FDU is that conventional FU is within a homogeneous (single-domain) setting, making focusing on the heterogeneous (multi-domain) setting more practical. However, the prior work \[1\] does not explicitly make such an assumption in its method. Besides, the prior work \[2\] even explicitly demonstrates that it uses non-IID data. Thus, the lack of a well-justified justification of previous studies’ assumption on the homogenous domain makes the focus on FDU less convincing and practical.

\[1\] Yi Liu, Lei Xu, Xingliang Yuan, Cong Wang, and Bo Li. The right to be forgotten in federated learning: An efficient realization with rapid retraining. In IEEE INFOCOM 2022-IEEE Conference on Computer Communications, pp. 1749–1758. IEEE, 2022.

\[2\] Junxiao Wang, Song Guo, Xin Xie, and Heng Qi. Federated unlearning via class-discriminative pruning. In Proceedings of the ACM Web Conference 2022, pp. 622–632, 2022.

C2: Assume previous studies only focus on homogeneous settings as the submission claims.

- First, when discussing the unique challenges brought by the heterogeneous setting compared to the homogeneous setting, the comparison should be within FDU (heterogeneous) and conventional FU (homogeneous). However, the submission claims the comparison is on “domain generalization,” a topic highly orthogonal to FDU.
- Second, the submission claims that current FU methods “demand substantially more resources and iterations to recover” in the heterogeneous setting. However, no solid evidence or justification supports such a claim.

C3:

- Please provide more solid evidence to justify why previous studies have only focused on homogeneous settings and demand more resources in heterogeneous settings.
- Why do previous methods “have weaker theoretical foundations and may introduce greater privacy vulnerabilities?” Any evidence or reference?

C4:

- The design relies on heuristic thresholds $\\tau_p$, $\\tau_f$ and assumes channel-wise independence. The pruning and freezing thresholds strongly influence performance, yet are selected empirically without a principled criterion. In addition, the method treats each channel independently, ignoring potential inter-channel correlations, which could cause partial forgetting or unintentional degradation of domain-invariant features.

---

> ### Author Response · Authors · 2025-11-28
> **Response to Reviewer iLq4 (1/4)**
>
> > Response to C1: Motivation of Federated Domain Unlearning)
>
> We thank the reviewer for raising this important clarification question. We agree that the notion of “heterogeneous” may have been misunderstood as *label-distribution non-IID*. In our paper, however, **“heterogeneous” strictly refers to domain heterogeneity rather than label heterogeneity**, which follows the standard formulation in **Federated Domain Learning (FDL)** (ref1, ref2). Specifically, in our considered setting, **each client belongs to a different domain, while all clients share the same label space**. This is fundamentally different from the commonly-studied non-IID label partition considered in conventional FL and Federated Unlearning (FU). We have further clarified this distinction in the revised manuscript. Accordingly, **Federated Domain Unlearning (FDU)** is *not* a hypothetical extension of FU, but arises naturally from real-world **FDL scenarios**, where an unlearning request involves **removing an entire domain** rather than a class or a small subset of samples. Following our formal definition:
>
> > *Given a model ( $f_\theta$ ) trained on ( $D_{train} = D_{retain} \cup {d_f}$ ), an unlearning request specifies a domain ( $d_f$ ). All clients belonging to ( $C_{d_f} $) withdraw, and the goal is to transform ( $f_\theta$) into ( $f_{\theta'}$ ) such that (i) the model behaves as if it had never seen ( $d_f$ ), and (ii) it preserves utility on the retained data ( $D_{retain}$ ).*
>
> This setting introduces **new challenges beyond existing FU works** (ref3):
>
> 1. **Label-space overlap across domains.**  In FDU, multiple domains can share identical labels. Therefore, domain-level forgetting cannot rely on label separation and is significantly more challenging than class-based unlearning.
>
> 2. **Feature-space shift is the primary difficulty.** While FU typically focuses on label-specific forgetting, FDU must remove *domain-specific representations* while preserving *label-specific knowledge*, due to large feature distribution differences between domains.
>
> 3. **Additional restoration objective.**   Unlike FU, where the primary goal is to recover performance on retained clients, FDU must also **restore domain generalization capability**, which is the original objective of FDL itself.
>
> The prior works mentioned by the reviewer (ref3) indeed consider non-IID label distributions, but they **do not address domain-based partitioning nor domain-level forgetting**, and thus cannot directly solve the FDU problem studied in our paper.
>
> To further avoid ambiguity, **we have revised Figure 2 and the corresponding text** to clearly illustrate
>
> (i) our cross-silo, domain-heterogeneous setting,
>
> (ii) the distinction from label-based non-IID FL/FU, and
>
> (iii) the complete logical chain from FDL → FDU.
>
> We believe these clarifications now make the motivation and necessity of FDU clearer, and further strengthen the practical relevance of our setting.
>
>
>
> **References**
>
> ref1: Wenke Huang, Mang Ye, Zekun Shi, He Li, and Bo Du. *Rethinking federated learning with domain shift: A prototype view.* CVPR 2023.
>
> ref2: Wenke Huang, Mang Ye, and Bo Du. *Learn from others and be yourself in heterogeneous federated learning.* CVPR 2022.
>
> ref3: Yi Liu et al., *The Right to Be Forgotten in Federated Learning*, IEEE INFOCOM 2022;
>
>     Junxiao Wang et al., *Federated Unlearning via Class-Discriminative Pruning*, WWW 2022.

---

> ### Author Response · Authors · 2025-11-28
> **Response to Reviewer iLq4 (2/4)**
>
> > Response to C2: On the relation between FDU and domain generalization.
>
> We agree that the comparison should be between **FDU (domain-heterogeneous)** and **conventional FU (domain-homogeneous)**. In our revision, we have clarified that “domain generalization” is *not* an orthogonal topic, but rather an **inherent objective of Federated Domain Learning (FDL)** (ref1, ref2), on top of which FDU is defined.
>
> In the FDL setting, each client belongs to a **single domain**, while all clients share the **same label space**. The goal of FDL is to achieve **domain generalization across heterogeneous domains**.
> Therefore, when a domain-level unlearning request is applied (i.e., removing domain ( $d_f$ )), the FDU objective naturally includes:
> (i) forgetting the contribution of ( $d_f$ ), and
> (ii) **restoring the domain-generalization capability** among the remaining domains.
> We have revised the text and Figure 2 to make this relationship explicit and avoid the impression that we compare unrelated objectives.
>
> ---
>
> >  Response to C2: On the claim of “larger resource and iteration cost” for existing FU methods.
>
> We now support this claim with **both structural analysis and empirical evidence**.
>
> **(a) Algorithmic mechanism & memory complexity**
>
> We compare a representative FU baseline, **Increase Loss (IL)** (ref3), with our **FU-DWS**.
>
> | Aspect                      | Increase Loss (IL)                                           | FU-DWS                                                       | Which is lighter?                         |
> | --------------------------- | ------------------------------------------------------------ | ------------------------------------------------------------ | ----------------------------------------- |
> | **Per-step extra objects**  | reference model θ_ref and unlearning model θ_u (both dense); one dense gradient buffer; random models only cached **during δ initialisation** | two small binary masks; one pruned model with active weights only | **FU-DWS**                                |
> | **GPU memory – asymptotic** | resident ≈ 2 w FP32; transient + 1 w buffer ⇒ **≈ (2 – 3) w bytes** | weights + optimizer states only for active fraction (1 – ρ) w; two bit-masks ρ w ⇒ **≈ (2 – 3)(1 – ρ) w bytes** | **FU-DWS** (≥ 1.3 × smaller when ρ ≤ 0.5) |
> | **Per PGD-step arithmetic** | compute ∇L and L₂-projection over w dims → **O(w)** ops      | mask gradients element-wise on active params → **O((1 – ρ) w)** ops | **FU-DWS**                                |
> | **Memory scaling**          | linear factor **2 – 3** (dense)                              | linear factor **(2 – 3)(1 – ρ)** (sparse)                    | **FU-DWS**                                |
> | **Initialisation overhead** | one-off cache of 10 random models to estimate δ              | no extra models                                              | **FU-DWS**                                |
>
> where ( w ) is the number of parameters and ( \rho ) is the pruning ratio.
>
> *A concrete example (ResNet-50, 25.6M parameters, ( $\rho = 0.3$ ))*:
> IL requires **205 MB** of persistent GPU memory (plus up to 102 MB transient), while our method requires only **≈133 MB**, resulting in a **~1.5× memory reduction**. This gap becomes larger for larger backbones (e.g., ViT-B/16).
>
> This explains, at a structural level, why existing dense-model FU methods are more resource-intensive.
>
> ---
>
> **(b) Empirical evidence on recovery speed (Table 2).**
>
> We further support the claim with data from **Office-Caltech-10 (Table 2 in the paper)**.
> Under domain-level unlearning, existing FU baselines typically require **2–40+ communication rounds** to recover the target retain accuracy, while **our method consistently achieves it in a single round (R# = 1)** across all cases.
>
> The relative convergence speedup of prior methods is only **0.03×–0.5×**, strongly demonstrating that existing methods are **significantly slower in recovery** when applied to the domain-heterogeneous FDU setting.

---

> ### Author Response · Authors · 2025-11-28
> **Response to Reviewer iLq4 (3/4)**
>
> >  Response to C3: Please justify why previous studies focus on homogeneous settings.
>
> We have revised the wording to avoid overgeneralization. Instead of stating that previous works “only” consider homogeneous settings, we now state more precisely that:
>
> Existing FU methods (ref3) primarily address **sample-, class-, or client-level deletion under standard FL assumptions**, including IID or label-based non-IID distributions, but **do not explicitly model domain-partitioned clients with shared label spaces**, as in Federated Domain Learning (FDL) (ref1, ref2).
>
> Thus, our work addresses a **different and underexplored unlearning scenario**: domain-level unlearning under domain shift, rather than class-level unlearning.
>
>
>
> >  Response to C3: On “weaker theoretical foundations” and “privacy vulnerabilities”.
>
> We agree that our original phrasing was too strong and could be misunderstood. Our intention was to distinguish between:
>
> * methods that rely on **heuristic optimization procedures**, and
> * methods that aim for **structured, domain-aware unlearning**.
>
> To avoid ambiguity, we have **removed the phrase “greater privacy vulnerabilities”** and revised the wording to:
>
> > “Most existing FU methods rely on heuristic optimization and do not provide guarantees specifically tailored to the domain-partitioned FDL setting considered in this work.”
>
> ---
> **References**
>
> ref1: Wenke Huang, Mang Ye, Zekun Shi, He Li, and Bo Du. *Rethinking federated learning with domain shift: A prototype view.* CVPR 2023.
>
> ref2: Wenke Huang, Mang Ye, and Bo Du. *Learn from others and be yourself in heterogeneous federated learning.* CVPR 2022.
>
> ref3: Yi Liu et al., *The Right to Be Forgotten in Federated Learning*, IEEE INFOCOM 2022;
>
>     Junxiao Wang et al., *Federated Unlearning via Class-Discriminative Pruning*, WWW 2022.

---

> ### Author Response · Authors · 2025-11-28
> **Response to Reviewer iLq4 (4/4)**
>
> > Response to C4:  On the use of heuristic thresholds.
>
> We acknowledge that the thresholds in FU-DWS are determined empirically. However, this choice is not arbitrary. Our thresholds are driven by **activation distribution statistics**, not hand-crafted rules, and are selected based on the **stability and separability of activation patterns** between the local client model and the global model. This design is consistent with existing practices in model compression and quantization literature (e.g., activation-aware methods such as AWQ, which demonstrate that activation magnitude reflects feature importance under different data distributions). Importantly, the threshold in FU-DWS does **not require access to the forget-domain data**. Instead, it is computed from the *retain client’s perspective* by measuring the discrepancy between the local and global activation responses. Units that exhibit significant divergence are interpreted as **domain-sensitive**, while activation-stable units are preserved via freezing. This provides a principled, distribution-driven mechanism rather than a purely heuristic choice. We have clarified this rationale in the revision.
>
>
>
>
>
> > Response to C4: On the assumption of channel-wise independence.
>
> FU-DWS does **not** assume strict channel independence. Instead, it adopts a **channel-level abstraction** as a practical intervention unit, while the *selection itself is activation-aware*, i.e., it is driven by **cross-client activation differences** between the local model and the global model. In other words, although pruning is applied on a per-channel basis, the **decision signal already embeds inter-channel and inter-feature correlations through network activations**, rather than treating channels as isolated or independent entities.
>
> To directly address the reviewer’s concern about correlated channels and partial overlap, we further conducted a **shared-domain multi-client experiment**, which creates explicit inter-client and inter-channel coupling:
>
> - We split each domain in **Office-Caltech-10** across **two clients** (8 clients in total).
> - An unlearning request was issued on **only one client** holding the target domain, while the other client of the *same domain* continued training.
> - We compared FU-DWS against the top-performing baseline **Increase Loss (IL)**.
>
> **Table R2. Results under shared-domain setting (each domain has 2 clients).**
>  *(Unlearning domain = Amazon. Higher Acc_↑ is better, lower Acc_F↓ and BKD↓ indicate stronger forgetting.)*
>
> | Method            | Acc_C ↑ | Acc_D ↑    | Acc_W ↑ | Acc_F ↓   | BKD ↓     |
> | ----------------- | ------- | ---------- | ------- | --------- | --------- |
> | RE                | 85.20   | 74.10      | 94.69   | 35.07     | 31.69     |
> | IL                | +7.22   | +6.30      | +4.47   | +10.86    | +9.88     |
> | **FU-DWS (Ours)** | 0.00    | **+10.36** | +2.04   | **+8.51** | **+7.15** |
>
> Despite domain overlap across clients, FU-DWS is able to **selectively suppress the influence of the forgetting client**, while preserving performance on retained domains. This demonstrates that:
>
> 1. FU-DWS remains effective even when **multiple clients share the same domain**;
> 2. It can isolate **fine-grained, client-specific domain signatures** through **activation-aware saliency**;
> 3. It avoids unintentional degradation of **domain-invariant features**, directly addressing the concern about correlated channels.
>
> These results indicate that channel-level intervention in FU-DWS is **not based on a naive independence assumption**, but rather guided by **implicit inter-channel correlation captured in the activation space**, which is sufficient to support reliable and fine-grained domain-level unlearning in practice.

---

### Author Response · Authors · 2025-12-01
**Latest updated content**

Thank you for carefully reading our paper and for your valuable suggestions. We have revised the manuscript according to your comments and highlighted the main changes in **blue** so that you can easily identify the modified content. We have also expanded the paper to a 10-page camera-ready version. **Specifically**, we have made the following updates:

**1**. We updated the original Figure 2 (now presented as Figure 1) and revised the corresponding descriptions. We further elaborated the design motivation of FU-DWS from four perspectives: Algorithm Aiming, Intuition, Observation, and Empirical Study.

**2**. We clarified the purpose of the FDU design and explained why it is necessary to consider generalization as a means to bridge the gap with traditional FUs.

**3**. We corrected unclear parts in the algorithm description to ensure reproducibility.

**4**. We added an analysis of the shared-domain setting.

**5**. We addressed the writing issues that you pointed out.

---

### Meta-Review · Area_Chair_ZNoF · 2026-01-07

**Summary:**

1. The paper's motivation is limited, as domain unlearning is technically very similar to client-wise unlearning.
2. The proposed method can be unstable and impractical, due to the reliance on heavy threshold tuning.
3. The experimental scope is narrow, limited to small image datasets.
4. Furthermore, the evaluation provides no formal unlearning guarantee
5. The proposed method can be problematic for entangled domains.

**Reviewer Concerns:**

Points 2, 4, and 5 were partially addressed. Others remain outstanding.

**Reviewer Scores:**

No change

---

### Decision · Program_Chairs · 2026-01-26

Reject